# Binary Oscillation - Regulated Network (BORN): approach for binary neural networks training

## Abstract

Binary Neural Networks (BNNs) are gaining attention for making energy-intensive deep learning more accessible to resource-constrained edge devices. Traditionally, training methods for such models focus on minimizing quantization error in forward propagation and approximating the sign function for gradient computation. However, these methods overlook the BNN-specific phenomenon of oscillations and do not adapt the learning rate to the training dynamics. To address this, we propose BORN, which adapts learning to oscillatory behavior. The proposal is based on two key innovations: an Oscillations-aware Sign Approximation, which reduces gradient information loss by gradually approximating the sign function in backward propagation, and an adaptive learning rate adjusted according to progress and oscillations. Beyond that, we are the first to combine a static sign approximation for activations with a dynamic one for weights, which stabilizes optimization while maintaining flexibility. Moreover, unlike prior work that chose approximations solely to replicate the sign function as closely as possible, we are the first to motivate the choice of approximating functions through the properties they induce for binary optimization, providing a principled foundation for their design. The novelty of BORN lies not only in its BNN-specific adaptive mechanisms but also in its ease of integration into existing architectures such as convolutional and transformer networks. To validate this, we integrate the method into several state-of-the-art (SoTA) BNN training frameworks across tasks like super-resolution (SR) and large language modeling (LLM). Experimental results show that the proposed method can cover an average of 34.0% of the proximity interval from SoTA methods to full-precision models.

## 1 Introduction

Deep neural networks (DNNs) have achieved SoTA performance across various domains (Tsirmpas et al., 2024; Archana & Jeevaraj, 2024), but at the cost of high computational and memory demands. Quantization addresses this by reducing the precision of weights and activations, improving storage and inference efficiency. Binarization, as an extreme form of quantization, uses one bit per parameter and enables significant speedups on resource-constrained hardware (Rastegari et al., 2016).

Research on BNNs (Yuan & Agaian, 2023) can be broadly categorized into Post-Training Quantization (PTQ) and Quantization-Aware Training (QAT). QAT integrates quantization into the training process, allowing the model to learn robust low-precision representations. We focus exclusively on QAT, as it enables the model to progressively adapt to binarization during training, preserving essential features and reducing accuracy degradation, whereas PTQ often leads to suboptimal solutions. QAT methods can be classified into five categories (Yuan & Agaian, 2023): *quantization error minimization* (Rastegari et al., 2016; Falkena et al., 2023; Zhou et al., 2016; Lu et al., 2023; Wang et al., 2019; Qin et al., 2020; Tu et al., 2022) – minimize information loss during binarization; *loss function improvement* (Liu et al., 2020; Xu et al., 2023) – add special distribution loss or regularization; *gradient approximation* (Courbariaux et al., 2016; Zareian et al., 2020; Liu et al., 2018; Qin et al., 2020; Qiu et al., 2022; Kim et al., 2020; Vargas et al., 2024) – improve backpropagation stability via approximate gradient to solve zero derivative issue; *network topology structure* (Zhang et al., 2022a; Liu et al., 2018; Phan et al., 2020; Bethge et al., 2021; Liu et al., 2020; Xu et al., 2022) – improve

and optimize BNN based on topology architecture; and *training strategies* (Liu et al., 2018; Qiu et al., 2022; Liu et al., 2020) – improve BNN performance through training strategy and tricks.

One of the major issues in BNN training is weight oscillation, wherein binary parameters repeatedly flip signs during optimization, preventing convergence to a stable solution (Xu et al., 2023). Oscillations in weight updates primarily stem from the discontinuous nature of the binarization function, which leads to high-gradient variance and instability in optimization (Xu et al., 2023).

Current approaches either use a fixed approximation throughout training (Bengio et al., 2013) or follow a predefined update rule (Qin et al., 2020) that lacks adaptability to the specific training dynamics of binary networks, which leads to the inaccurate gradient updates.

Another limitation in BNN training is the lack of a learning rate (LR) scheduler specifically designed for binary networks (Yuan & Agaian, 2023). Standard LR schedulers, such as StepLR or CosineAnnealingLR (Loshchilov & Hutter, 2016), don't account for the unique challenges of binarized weight updates, where abrupt sign changes and high-gradient variance can destabilize training.

**Our contribution** We propose *Binary Oscillation-Regulated Network (BORN)*, a novel algorithm for training BNNs that mitigates oscillation-related instability and accelerates training convergence. The core of BORN is an adaptive approximation of the sign function for weights, which makes use of the epoch number and the current oscillation rate to provide a stable gradient dynamics. Additionally, we introduce a static sign function approximation for activations, which allows for control over the magnitude of the backpropagated gradients, thereby improving convergence efficiency for the weight groups that produce activations whose value is close to zero. *BORN is the first to combine adaptive and static* sign *approximations within a unified hybrid framework.* A third component is a dynamic LR scheduler extension, which adapts LR values based on the type of sign approximation in use. We also provide a theoretical convergence analysis of BORN, proving the convergence guarantees. Furthermore, we analytically show that BORN achieves a lower final loss compared to the Straight-Through Estimator (STE) (Courbariaux et al., 2016). The convergence rates are assessed within the general theoretical framework of Li et al. (2017). For empirical benchmarking, we compare BORN against the top-performing methods from five major QAT algorithm categories (Yuan & Agaian, 2023); the selection rationale is detailed in Section 2. Experimental results demonstrates a mean improvement 34.0% across models, with per-architecture averages of 42.99% for EDSR, 26.21% for SRResNet, 37.02% for GPT-2 and 29.81% for ResNet-18.

## 2 RELATED WORK

**Quantization-aware training** Gradient approximation for training BNNs started with BinaryConnect (Courbariaux et al., 2015). BiReal-Net (Liu et al., 2018) uses a piecewise polynomial to approximate the sign function. IR-Net (Qin et al., 2020) considers binarization from the information flow perspective, adds weight standardization and a dynamic gradient estimator to reduce approximation errors. Quantization error reduction methods aim to keep information when binarizing. XNOR-Net (Rastegari et al., 2016) uses per-channel scaling for weights and activations. AdaBin (Tu et al., 2022) learns the optimal binary values per layer. LAB (Falkena et al., 2023) replaces the sign function with a learnable binarizer. Bop (Helwegen et al., 2019) handles binary weights only, flipping them based on the accumulated gradients. Bop2ndOrder (Suarez-Ramirez et al., 2021) adds second-moment estimation to Bop, following Adam (Kingma, 2014). BiPer (Vargas et al., 2024) uses periodic square waves for binarization and sine waves for gradient approximation. VISPA (Orecchia et al., 2024) applies Gaussian variational inference for BNN training. Some methods improve training by changing the loss or network structure. ReActNet (Liu et al., 2020) reshapes activation distributions and adds a distributional loss. ReBNN (Xu et al., 2023) studies weights oscillations in BNNs and adds a modified loss with a scaling factor. In image SR, BBCU (Xia et al., 2022) designs a binary conv unit that removes batch normalization. E2FIF (Song et al., 2023) improves BNNs by adding FP information flow for stronger forward and backward signals.

Most BNN methods were made for specific tasks like classification (CLF) or SR, which limits generalization. Common issues include information loss caused by binarization (Courbariaux et al., 2015), vanishing gradients from sign approximations (Liu et al., 2018), and the introduction of large number of additional learnable parameters (Orecchia et al., 2024). To compare binarization methods, we choose top-performing baselines from each method group (Yuan & Agaian, 2023). For

CLF (ResNet-18 (He et al., 2016a)), we include ReActNet (Liu et al., 2020), LAB (Falkena et al., 2023), AdaBin (Tu et al., 2022) because of strong performance on ImageNet (Deng et al., 2009) (see Table 5); for SR (SRResNet), we include BBCU (Xia et al., 2022) (see Table 6). We also include BNN (STE) (Courbariaux et al., 2016) as a fundamental algorithm to validate theoretical improvements, and IR-Net (Qin et al., 2020) for its use of a gradient approximation strategy conceptually related to ours, yet distinct in formulation and application. We restrict our comparison to peer-reviewed and published works, and do not include preprints, in order to ensure the stability and reproducibility of the reported baselines.

**Theoretical studies of BNNs** Theoretical studies on DNNs covers: *capacity* (Guss & Salakhutdinov, 2018; Scherlis et al., 2022), *training/convergence* (Gower et al., 2019; Zhang et al., 2022b), *generalization* (Allen-Zhu et al., 2019), and *robustness* (Webb et al., 2018). For BNNs, most studies focus on convergence rates (Li et al., 2017). STE is a key method in quantized network training. Yin et al. (2019) show that STE can cause divergence, but also give convergence guarantees. Limits of work by Yin et al. (2019) include focusing on 2-layer networks and complex calculations.

**Adaptive learning rate** Adaptive LR methods are of two types: *gradient-based methods* (e.g., Adadelta (Zeiler, 2012), Learnable LR (Xu et al., 2019)), *loss-based methods* (e.g., Loss Based LR (Behera et al., 2006)). Gradient-based methods do not transfer well to BNNs due to discontinuities, which lead to unstable LR values. Loss-based methods show significant performance improve only while being used on FP networks – BNNs have complex loss landscapes that cause LR to fluctuate, making the training process unstable. Also it is important to mention that none of the existing approaches can be reformulated in binary-specific way that exploits binarization tricks used in QAT.

## 3 BINARY OSCILLATION-REGULATED NETWORK

This section presents BORN, a novel algorithm for BNN training that requires no architectural modifications of a base model to be binarized. Section 3.1 outlines training scheme of the BORN. Then, in Sections 3.2–3.4 we expose each part of the algorithm. Specifically, we examine how oscillations impact BNN training and introduce an oscillation-regulated sign approximation, explore the role of activations distribution in BNN training and present a sign approximation for activations, propose a BNN-specific LR adjustment.

### 3.1 TRAINING SCHEME OF BORN

**Forward propagation** The output of a layer in DNN with weights vector $\boldsymbol{w} \in \mathbb{R}^d$ and activations $\boldsymbol{a} \in \mathbb{R}^d$ can be expressed by $y = \boldsymbol{w}^T \boldsymbol{a}$. We define binarization functions for weights and activations as $Q_w(\boldsymbol{w}) = \frac{\|\boldsymbol{w}\|_1}{d} \operatorname{sign}(\boldsymbol{w})$, $Q_a(\boldsymbol{a}) = \operatorname{sign}(\boldsymbol{a})$, where sign is applied element-wise. The factor $\frac{\|\boldsymbol{w}\|_1}{d}$ (that is calculated per channel) allows the binarized weights $\operatorname{sign}(\boldsymbol{w})$ to minimize the quantization error (Rastegari et al., 2016).

Thus, we formulate the output of a layer with binarized weights and activations as $y = Q_w(\boldsymbol{w})^T Q_a(\boldsymbol{a}) = \frac{\|\boldsymbol{w}\|_1}{d} \left( \operatorname{sign}(\boldsymbol{w})^T \operatorname{sign}(\boldsymbol{a}) \right)$.

**Backward propagation** Since the derivative of sign function is zero almost everywhere, its straightforward usage is incompatible with backward propagation algorithm. This leads to the introduction of gradient approximation techniques for training BNNs that can be expressed as $\frac{\partial L}{\partial \boldsymbol{w}} = \frac{\partial L}{\partial Q_w(\boldsymbol{w})} \frac{\partial Q_w(\boldsymbol{w})}{\partial \boldsymbol{w}} \approx \frac{\partial L}{\partial Q_w(\boldsymbol{w})} g'(\boldsymbol{w})$ and $\frac{\partial L}{\partial \boldsymbol{a}} = \frac{\partial L}{\partial Q_a(\boldsymbol{a})} \frac{\partial Q_a(\boldsymbol{a})}{\partial \boldsymbol{a}} \approx \frac{\partial L}{\partial Q_a(\boldsymbol{a})} h'(\boldsymbol{a})$, where $L$ denotes the loss function, $Q_w, Q_a$ represent the weight and activation binarization functions, correspondingly, and $g, h$ are the approximation of $Q_w, Q_a$, correspondingly.

Approximation methods for BNN training are typically classified as static (Courbariaux et al., 2015; Liu et al., 2018) or dynamic (Qin et al., 2020; Lu et al., 2023; Cai et al., 2023). Static approaches lack adaptability; dynamic ones—dynamic only during training, not the training situation—may suffer from information loss over time due to a reduced update range for large activations or weights.

To address these limitations, we introduce Oscillations-aware Sign Approximation (**OSA**): $g(x) = k \tanh(tx)$, and Static Sign Approximation (**SSA**): $h(x) = \frac{F}{b} \tanh(bx)$, where $k, t \in \mathbb{R}$ are oscillation-driven parameters and $F, b \in \mathbb{R}$ are hyperparameters. A detailed explanation of the $g$ and $h$ update process and LR adjustment will be provided in Sections 3.2–3.4.

### 3.2 OSCILLATIONS-AWARE SIGN APPROXIMATION

BORN leverages the concept of oscillations, as discussed by Xu et al. (2023). The oscillation of a binary weight $w_b$ at the $i$-th iteration is defined as the change in its value over the three preceding iterations: $\text{osc}(w_b, i) = \left( w_b^{i-2} \oplus w_b^{i-1} \right) \wedge \left( w_b^{i-1} \oplus w_b^i \right)$, where $\oplus$ and $\wedge$ denote the XOR and AND logical operations, respectively. The total oscillation at iteration $i$ is given by $\text{osc}(i) = \sum_{w_b} \text{osc}(w_b, i)$.

Oscillations can be interpreted in two distinct ways. Firstly, oscillations hinder weight convergence, particularly if they persist towards the later stages of training. In this case oscillatory weights don't contribute to further learning, necessitating an artificial intervention to promote convergence. Secondly, oscillations signify the exploration of a cluster of local minima, a characteristic feature of the loss landscape in BNNs (Li et al., 2018). In this case, oscillations suggest that the training is navigating a region of local minima in search of an optimal weight vector. Since prolonged stagnation within such clusters is undesirable due to the potential existence of superior minima of loss function, it is essential to facilitate further training after an initial exploration phase.

To effectively address both cases, we propose the following Oscillations-aware Sign Approximation:

$$g(x) = k(i, n) \tanh\left(t(i, n)x\right), \quad g'(x) = k(i, n)t(i, n)\left(1 - \tanh^2(t(i, n)x)\right) \quad (1)$$

where $t, k$ are dynamically adjusted according to

$$t(i, n) = T_{\min} \times 10^{(\lambda_1 t_{\text{epoch}}(n) + \lambda_2(1 - t_{\text{osc}}(i, n))) \times \log_{10} \frac{T_{\max}}{T_{\min}}}, \quad k(i, n) = \max\left(\frac{1}{t(i, n)}, 1\right), \quad (2)$$

$$t_{\text{epoch}}(n) = \frac{n}{N}, \quad t_{\text{osc}}(i, n) = \frac{n}{N}\frac{\text{osc}(i)}{\text{osc}_{\max}(n)}, \quad \text{osc}_{\max}(n) = \max_{\text{iter } j \text{ in } (n-1)\text{-th epoch}} \text{osc}(j), \quad (3)$$

where $i$ is the iteration number, $j$ is the iteration number for the previous epoch, $n$ is the current epoch index, and $N$ is the total number of epochs. $\lambda_1$, $\lambda_2$ serve as scaling constants for training progress and oscillation sensitivity, respectively. $T_{\min}$ and $T_{\max}$ (recommended values are $1/10$, $10$, correspondingly) define the lower and upper bounds for $t$. Oscillations are computed over the entire binarized part of the network.

OSA has a key property: it adaptively stabilizes weights that have already settled in their sign during training with decrease of gradient's value for weights with large absolute value, while simultaneously correcting oscillatory weights that are close to zero with increasing gradient's value (see Fig 1). Adaptability occurs due to the fact that the sub-components $t_{epoch}$ and $t_{osc}$ allow you to vary $t$ for $g$. $t_{epoch}$ increases over time (increasing $t$) to approximate the sign by the end of the cut. $t_{osc}$ varies depending on the learning situation: when the oscillations jump, $t_{osc}$ increases (decreasing $t$), allowing the modulo weights to be updated to move to the next cluster of local minima, as well as reducing gradient for weights nearby zero, which prevents convergence.

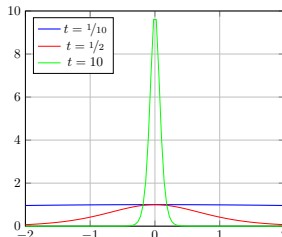

Figure 1: Evolution of $g'$ with respect to $t$.

Parameters $\lambda_1$ and $\lambda_2$ control the influence of training progress and oscillations on weight approximation. It is preferable to avoid setting $\lambda_1$ too low (recommended value is $\lambda_1 = 1$), as this would prevent the OSA from getting close to sign function (in terms of $L_2$ norm) towards the end of training. As for $\lambda_2$, selecting a very large value (recommended value is $\lambda_2 = 1/8$) may cause the large gradients for weights near zero, thereby reducing their ability to settle on a definitive sign in the end of training. A smaller value $\lambda_2$ lets the training to update larger weights for a longer period of time, while allowing to react to oscillations in approximately the same way as with a larger $\lambda_2$ value. Sensitive analysis and advice on choosing values can be found in F.3.

### 3.3 STATIC SIGN APPROXIMATION

In BORN, we employ a Static Sign Approximation:

$$h(x) = \frac{F}{b}\tanh(bx), \qquad h'(x) = F\left(1 - \tanh^2(bx)\right) \quad (4)$$

where $F$ and $b$ are hyperparameters governing the shape and behavior of the approximation. We use SSA exclusively to activations, thereby preserving non-vanishing gradients for groups with near-zero activations and accelerating convergence of the corresponding weights, in contrast to time-adaptive approximation methods, which remain prone to gradient vanishing.

$F$ (recommended value is 2.5) determines the maximum value of $h'$. In contrast, $b$ (recommended value is 1) controls the range of nonzero gradients (see Fig 2). The choice of relevant values for these parameters depends on the complexity of the backpropagation process, which is influenced by architectural and methodological factors that impact activations distribution and gradient's behavior, including: *network topology* (Vaswani et al., 2017; Krizhevsky et al., 2012), *architectural components* (He et al., 2016b; Falkena et al., 2023; Lin et al., 2015; Ioffe & Szegedy, 2015), *activation functions*, *regularization techniques* (Srivastava et al., 2014; Krogh & Hertz, 1991).

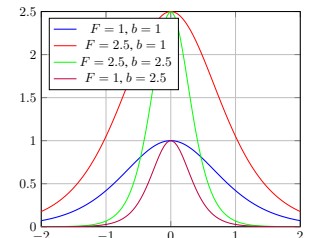

Figure 2: Visualization of $h'$ for different combinations of $F, b$.

Advices on choosing values of $F$ and $b$ according to the above factors can be found in F.2.

### 3.4 ADAPTIVE LEARNING RATE

To make LR adaptive to the current approximation state, it can be observed that the lower the value of $t$ is, the faster should be the growth of LR value to allow compensating the unstable gradient dynamics. Hence, to provide an adaptive extension for any LR scheduler, it was decided to add a value based on the current OSA state to make the LR adjustable to the evolution of the OSA. However, the added values should be bounded to avoid causing significant yet redundant growth of LR. We formulate Adaptive Learning Rate the following way:

$$\gamma_n = \left[ \overline{\gamma}_n + \frac{T_{\max} - t(m, n-1)}{T_{\max} - T_{\min}} \times \eta \right]_{\gamma_{\min}}^{\gamma_{\max}}, \quad [x]_a^b = \min\{b, \max\{a, x\}\} \tag{5}$$

Here $[T_{\min}; T_{\max}]$ is the range for $t$, $t(m, n-1)$ is value of $t$ for the previous epoch $n-1$ and its last iteration $m$, $\overline{\gamma}_n$ is the basic LR value (one obtained from the scheduler being extended), $\eta$ is the addition weight factor (denoting the overall influence of the modification; recommended value $\eta = \gamma_{min}$, which makes the modified curve reasonably close to the baseline scheduler curve), $\gamma_{\min}$ and $\gamma_{\max}$ are the lowest and the highest possible LR values, respectively. An example of such adaptive modification is presented on Fig. 3. The recommendations on choosing the values for the Adaptive Learning Rate parameters are presented in F.4.

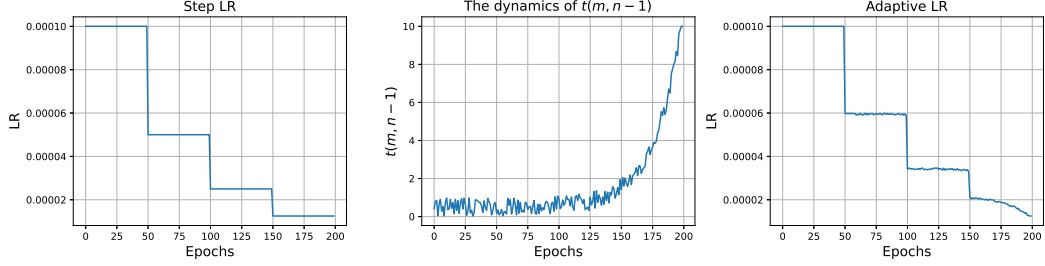

Figure 3: An example of Adaptive LR modification. The left figure shows the base Step LR schedule. The middle figure depicts the dynamics of $t$ as a function of epochs and iterations. The right figure presents the resulting LR schedule after applying the modification based on the formula (5).

Algorithm 1 summarizes the BORN training loop, which performs forward binarization, updates OSA, Adaptive LR, and applies weight updates based on the approximated gradients.

---

**Algorithm 1** BNN training using BORN

---

**Input:** Loss function $L$, initialized FP weights of considered layer $\boldsymbol{w} \in \mathbb{R}^d$, LR values $\overline{\gamma}_k$, min and max LR values $\gamma_{\min}$ and $\gamma_{\max}$, respectively, number of iterations $m$ per epoch, number of epochs $N$, hyperparameters $F$ and $b$ for SSA, $T_{\min}, T_{\max}, \lambda_1, \lambda_2$ for OSA, LR modification parameter $\eta$

**Require:** Input data $\boldsymbol{a} \in \mathbb{R}^d$, FP weights $\boldsymbol{w} \in \mathbb{R}^d$, current iteration and epoch numbers $i, n$

1: **while** number of epochs less than $N$ **do**

2: $\quad y = \frac{\|\boldsymbol{w}\|_1}{d} \left(\text{sign}(\boldsymbol{w})^T \text{sign}(\boldsymbol{a})\right)$; $\qquad\qquad\qquad\qquad$ ▷ Calculate the output of layer

3: $\quad g(x) = k(i, n) \tanh{(t(i, n)x)}$; $\qquad\qquad\qquad$ ▷ Update $k$ and $t$ of OSA by equation (2)

4: $\quad \gamma_n = \left[\overline{\gamma}_n + \frac{T_{\max} - t(m, n-1)}{T_{\max} - T_{\min}} \times \eta\right]_{\gamma_{\min}}^{\gamma_{\max}}$; $\qquad\qquad$ ▷ Update Adaptive LR by equation (5)

5: $\quad \frac{\partial L}{\partial \boldsymbol{a}} = \frac{\partial L}{\partial Q_a(\boldsymbol{a})} h'(\boldsymbol{a}), \frac{\partial L}{\partial \boldsymbol{w}} = \frac{\partial L}{\partial Q_w(\boldsymbol{w})} g'(\boldsymbol{w})$; $\quad$ ▷ Calculate $\nabla L$ using sign approximations

$\quad$ $g$ (OSA) and $h$ (SSA) by equations (1), (4)

6: $\quad \boldsymbol{w} \leftarrow \boldsymbol{w} - \gamma_n \frac{\partial L}{\partial \boldsymbol{w}}$; $\qquad\qquad\qquad\qquad\qquad$ ▷ Update the parameters of the model

7: **end while**

---

## 4 THEORETICAL STUDY OF BORN

This section analyzes the convergence of BORN. In Section 4.1, we prove its convergence and compare its final loss value to that of STE to prove the efficiency of the approximation that is refined over time. Then, in Section 4.2, we apply the method from (Li et al., 2017) to estimate its convergence rate. Notation, supporting results, and full proofs can be found in Appendix B. We also include the convergence rate of STE there as an extra result.

### 4.1 CONVERGENCE OF BORN

We provide a theoretical guarantee for the convergence of BORN weights to the optimal binary weights. Consider minimizing a smooth loss $L : \mathbb{R}^d \to \mathbb{R}$ under binary constraints:

$$\boldsymbol{w}_b^* \in \arg\min_{\boldsymbol{w} \in \{-1,1\}^d} L(\boldsymbol{w}). \tag{6}$$

This problem cannot be solved with gradient methods. Most approaches use a *pseudo-gradient* idea. Let $\boldsymbol{w} = (w_1, \ldots, w_d)^T$ and define the binary loss as $L_b(\boldsymbol{w}) = L(\text{sign}(w_1), \ldots, \text{sign}(w_d))$. Since the derivative of sign is zero almost everywhere, $\nabla L_b$ cannot be used for backpropagation. BORN handles this problem by replacing the derivative of sign with $g_\alpha'$, where $\alpha \in \mathbb{R}$ controls how closely $g_\alpha$ approximates sign. A larger $\alpha$ gives a more accurate approximation.

**Assumption 1.** Let $\varepsilon > 0$ be a given small constant and $\bar{\alpha}$ be such that

$$\|g_{\bar{\alpha}} - \text{sign}\|_{L_2(\mathbb{R})} \leq \varepsilon. \tag{7}$$

Then, to an accuracy of $\varepsilon$ we can assume that $L_{\bar{\alpha}}(\boldsymbol{w}) = L(g_{\bar{\alpha}}(w_1), \ldots, g_{\bar{\alpha}}(w_d)) \approx L_b(\boldsymbol{w})$.

The change of $\alpha$ in BORN starts from the value $\alpha_0$ that coincides with the parameter value of the approximation $g_{\alpha_0}$ employed by STE, so we suppose that $\alpha \in [\alpha_0, \bar{\alpha}]$. From now on, under the Assumption 1, we treat $L_{\bar{\alpha}}$ and $L_b$ as equal.

**Definition 1.** Pseudo-gradient of $L_b$ with parameter $\alpha$ is a vector that is denoted as $\nabla_\alpha L_b$ and equals to $\nabla_\alpha L_b(\boldsymbol{w}) = \left(\partial_1 L\left(\text{sign}(w_1)\right) g_\alpha'(w_1), \ldots, \partial_d L\left(\text{sign}(w_d)\right) g_\alpha'(w_d)\right)^T$, where $\partial_i L$ denotes the partial derivative of $L$ with respect to the $i$-th variable.

Gradient descent with pseudo-gradient $\nabla_\alpha L_b$ and step size $\gamma$ for problem (6) on the iteration $t$ can be written as:

$$\boldsymbol{w}^t = \boldsymbol{w}^{t-1} - \gamma \nabla_\alpha L_b\left(\boldsymbol{w}^{t-1}\right). \tag{8}$$

Process (8) is actually a discrete analog of the following continuous model:

$$\frac{d\boldsymbol{w}}{dt}(t) = -\nabla_\alpha L_b\left(\boldsymbol{w}(t)\right), \tag{9}$$

since the Euler's method for numerical solving the system of ordinary differential equations (9) coincides with (8).

**Proposition 1.** *Under the conditions of Assumption 1, to an accuracy of $\varepsilon$ we can assume, that* $\nabla_{\bar{\alpha}} L_b(\boldsymbol{w}) \approx \nabla_{\boldsymbol{w}} L_{\bar{\alpha}}(\boldsymbol{w})$.

Therefore, by Proposition 1, we use the stationary point $\boldsymbol{w}_{\bar{\alpha}}^*$ of (9) with $\alpha = \bar{\alpha}$ as the minimizer of $L_b$.

**Assumption 2.** Assume that the binary loss $L_b$ is convex in some neighbourhood of $\boldsymbol{w}_{\bar{\alpha}}^*$.

Since the equality of the gradient to zero at some point for the convex function is a necessary and sufficient condition for this point to be the minimum, then, under the Assumption 2, we treat the obtained point $\boldsymbol{w}_{\bar{\alpha}}^*$ as a minimum of $L_b$.

**Theorem 1** (Convergence of BORN). *Let the right-hand side of (9) satisfy the conditions of theorem on the continuous dependence of the solution of system of ordinary differential equations on the parameter (Artstein, 1975). Let $\alpha_k \xrightarrow[k \to +\infty]{} \bar{\alpha}$, then the sequence of stationary points $\boldsymbol{w}_{\alpha_k}^*$ of (9) with $\alpha = \alpha_k$ converges to $\boldsymbol{w}_{\bar{\alpha}}^*$.*

*Remark* 1. The statement about obtaining the stationary point $\boldsymbol{w}_{\alpha_k}^*$ for each $k$ matches the actual training process. In fact, at each epoch we are dealing with the fixed value of $\alpha$. Therefore, without loss of generality, we can assume that the $k$-th epoch contains a sufficient number of iterations to obtain the stationary point $\boldsymbol{w}_{\alpha_k}^*$.

Theorem 1 shows that with the enhancement of the approximation of sign the optimal solution of the approximation problem converges the optimal solution of the binary problem. This is *exactly* the idea behind the method of training binary networks using an approximation refined over time, including BORN.

**Corollary 1** (STE solution quality). STE generates a solution $\boldsymbol{w}_{\alpha_0}^*$, where $\alpha_0 \leq \bar{\alpha}$, such that $L_b(\boldsymbol{w}_{\bar{\alpha}}^*) \leq L_b(\boldsymbol{w}_{\alpha_0}^*)$.

*Remark* 2. It is obvious that the binary loss $L_b$ will not necessarily decrease on the solutions of (9) with $\alpha = \alpha_0$, which corresponds to STE. To back up this argument, one can obtain $\frac{d}{dt} L_{\bar{\alpha}}(\boldsymbol{w}(t)) = \nabla L_{\bar{\alpha}}(\boldsymbol{w}(t)) \frac{d\boldsymbol{w}}{dt}^T(t) \stackrel{(9)}{=} -\nabla L_{\bar{\alpha}}(\boldsymbol{w}(t)) \nabla_{\alpha_0} L_b(\boldsymbol{w}(t))^T$, which can be positive. However, $\frac{d}{dt} L_{\bar{\alpha}}(\boldsymbol{w}(t)) \stackrel{(9)}{=} -\nabla L_{\bar{\alpha}}(\boldsymbol{w}(t)) \nabla L_{\bar{\alpha}}(\boldsymbol{w}(t))^T = -\|\nabla L_{\bar{\alpha}}(\boldsymbol{w}(t))\|^2 \leq 0$, meaning that the binary loss decreases on the solution of (9) with $\alpha = \bar{\alpha}$.

## 4.2 CONVERGENCE RATE ESTIMATE OF BORN

In this section, we analyze the convergence rate for a BNN training using BORN. We consider a fully connected network with one hidden layer of $l$ ReLU ($f(x) = \max\{0, x\}$) neurons, trained on a regression task using squared error loss: $L(\boldsymbol{Y}, \hat{\boldsymbol{Y}}) = \sum_{i=1}^{N} \|\boldsymbol{y}_i - \hat{\boldsymbol{y}}_i\|^2$, where $\boldsymbol{y}, \hat{\boldsymbol{y}}$ are $N \times m$ matrices with columns $\boldsymbol{y}_i, \hat{\boldsymbol{y}}_i$, correspondingly. Without loss of generality, let $(\boldsymbol{x}_i, \boldsymbol{y}_i) \in (0; 1)^n \times (0; 1)^m$ be the input-output pair. Denote the predicted output for input $\boldsymbol{x}_i \in (0, 1)^n$ as $\hat{\boldsymbol{y}}_i \in \mathbb{R}^m$.

We study stochastic gradient descent (SGD), where at each step $t$ a random data point $\boldsymbol{x}_i$ is picked and weights are updated using $\boldsymbol{w}^t = \boldsymbol{w}^{t-1} - \alpha_t \nabla F_i(\boldsymbol{w}^{t-1})$, with $F_i = \|\boldsymbol{y}_i - \hat{\boldsymbol{y}}_i\|^2$, and $\nabla F_i$ is approximated using with OSA and SSA sign approximations. From now on we drop the index $i$ in $F_i$, $\boldsymbol{x}_i$, $\boldsymbol{y}_i$, $\hat{\boldsymbol{y}}_i$ for the convenience.

Assume all weights are bounded by a constant $\Omega$. Then, predictions $\hat{\boldsymbol{y}}_i$ are also bounded.

**Lemma 1.** *If BORN is used, then gradient norms are bounded: $\|\nabla F\|^2 \leq G^2$, where $G^2$ is a constant depending on network size, prediction error bound $H$, and $\Omega$.*

**Lemma 2.** *The norm of the quantization error from weight binarization is also bounded: $\|\mathbf{r}\|^2 \leq R^2$, with $R^2 = 2l(m+n)\Omega^2$.*

**Theorem 2** (BORN Convergence Rate). *Let $\{\boldsymbol{w}^\tau\}$ be the sequence of weights generated by SGD with BORN and learning rates $\{\gamma_\tau\}$ such that $\boldsymbol{w}^\tau \xrightarrow[\tau \to \infty]{} \boldsymbol{w}^*$. Assume the loss $L$ is convex and its gradient is Lipschitz continuous with constant $\mathcal{L}$. Then $\mathbb{E}(L(\overline{\boldsymbol{w}}^T) - L(\boldsymbol{w}^*)) \leq \frac{4l(m+n)\Omega^2}{T\gamma_1} + \frac{4l(m+n)\Omega^2}{T}\left(\frac{1}{2\gamma_T} - \frac{1}{2\gamma_1}\right) + \frac{G^2}{2T}\sum_{\tau=1}^{T}\gamma_\tau + 2\mathcal{L}R^2 l(m+n)\Omega^2$, where $\overline{\boldsymbol{w}}^T = \frac{1}{T}\sum_{\tau=1}^{T}\boldsymbol{w}^\tau$.*

The two theoretical results address complementary aspects of convergence. Theorem 1 establishes convergence to the optimal binary solution from an ODE-based perspective, which is a novel approach in BNN theory. As a corollary, static approximations such as STE may fail to converge optimally, emphasizing the need for progressively refined approximations. Theorem 2 complements this by providing a convergence rate estimate, allowing direct comparisons between different training methods. Together, these results provide both qualitative and quantitative insights into the convergence behavior of BORN. In Appendix B.11, the validity of Theorem 2 is shown in the experiment with the same setting of fully connected network.

## 5 EXPERIMENTS, ANALYSIS AND DISCUSSIONS

### 5.1 EXPERIMENTAL SETTINGS

**Datasets and neural network architectures description** We trained SRResNet (Ledig et al., 2017) and EDSR (Lim et al., 2017) models on the DIV2K (Agustsson & Timofte, 2017) dataset, evaluating them on Set5 (Bevilacqua et al., 2012) and Set14 (Zeyde et al., 2010) benchmarks. Experiments covered standard and modified SRResNet and EDSR architectures, incorporating adjustments from recent binarization studies. Analogously, we evaluated ResNet-18 on CIFAR-10 (Krizhevsky, 2009) and ImageNet (Deng et al., 2009). To assess the generalizability of our binarization approach, we adapted the GPT-2 model from the LLM domain, testing it on binary sentiment classification (IMDB dataset (Maas et al., 2011)) and text generation (WikiText-2 (Merity et al., 2016)). Experimental hyperparameters can be found in Appendix D. The rationale for choosing GPT-2 can be found in Appendix H.

**Quality metrics of BNN training** For quantifying image quality, we employed two standard metrics – Peak Signal-to-Noise Ratio (PSNR) and Structural Similarity Index Measure (SSIM). These metrics were calculated exclusively on the luminance (y) channel, as this component is most closely aligned with human visual perception. We evaluated language models using Accuracy for classification tasks and Perplexity for text generation tasks. To enable a more refined and comprehensive comparison of the models under investigation, we introduced three custom-designed metrics that are tailored to capture aspects of model performance not fully addressed by conventional metrics. A detailed description of the custom quality metrics ($\mu_q$ – FP Difference, $\mu_m$ – Relative Memory Usage, $\mu_p$ – Convergence Plateau Comparison) and BORN custom metrics values are provided in Appendix E.

### 5.2 ABLATION STUDY

We further investigate the performance using different parts of BORN with EDSR model on DIV2K Set5 and Set14; ResNet-18 on CIFAR-10. Table 1 contains such analysis for IR-Net (Qin et al., 2020) and STE baselines. It is evident that each part of the method plays an important role, moreover, the improvements made by each individual part are superimposed, which underlines the importance of their interconnection. Table 9 in Appendix D summarizes the hyperparameters used in our experiments. Other evaluation results for classification outside the ablation study can be found in 3.

Table 1: Ablation study of BORN for SR and CLF tasks on EDSR and ResNet-18 (with hardtanh activation function) architectures.

| | EDSR (x4 scale) | | | | ResNet-18 | | |
| | Set5 | | Set14 | | CIFAR-10 | Imagenet | |
| Method | PSNR | SSIM | PSNR | SSIM | Acc.(%) | Top-1(%) | Top-5(%) |
|---|---|---|---|---|---|---|---|
| FP | 31.421 | 0.895 | 27.819 | 0.789 | 92.64 | 69.6 | 89.2 |
| IR-Net (Qin et al., 2020) | 21.761 | 0.588 | 21.104 | 0.519 | 91.49 | 58.1 | 80.0 |
| STE | 28.46 | 0.81 | 26.03 | 0.71 | 90.52 | 27.9 | 50.42 |
| BORN (our) | **30.090** | **0.867** | **27.005** | **0.766** | **91.87** | **58.3** | **80.24** |
| BORN (wihtout SSA & ALR) | 29.245 | 0.829 | 26.638 | 0.729 | 91.53 | 58.2 | 80.11 |
| BORN (without SSA) | 29.257 | 0.829 | 26.646 | 0.729 | 91.58 | 58.2 | 80.16 |
| BORN (without ALR) | 29.303 | 0.831 | 26.674 | 0.730 | 91.76 | 58.3 | 80.21 |
| BORN (without OSA & ALR) | 29.239 | 0.828 | 26.635 | 0.728 | 90.36 | 58.1 | 80.09 |
| BORN (without OSA & SSA) | 29.237 | 0.828 | 26.632 | 0.728 | 90.33 | 58.1 | 80.04 |

## 5.3 COMPARISON WITH SoTA METHODS

BORN demonstrates a mean $\mu_q$ improvement of 34.0% across models, with per-architecture averages of 42.99% for EDSR, 26.21% for SRResNet, 37.02% for GPT-2 and 29.81% for ResNet-18 (see Tables 11, 12, 13, 14). The best improvement is achieved by EDSR (IR-Net) 89.11%, indicating strong quality retention in SR tasks. A description of the custom metrics is provided in Table 10.

Table 3: ResNet-18 classification benchmarks.

| Method | Imagenet | | CIFAR-10 |
|---|---|---|---|
| | Top-1 | Top-5 | Top-1 |
| | Acc.(%) | Acc.(%) | Acc.(%) |
| FP | 69.6 | 89.2 | 92.64 |
| IR-Net | 58.1 | 80.0 | 91.49 |
| STE | 27.9 | 50.42 | 90.52 |
| BORN[1] | **58.3** | **80.24** | **91.87** |
| LAB | 64.2 | 85.0 | 84.1 |
| BORN[2] | **64.26** | **85.2** | **85.0** |
| ReActNet | 65.5 | 86.1 | 92.31 |
| BORN[4] | **65.6** | **86.2** | **92.5** |
| AdaBin | 66.4 | 86.5 | 93.1 |
| BORN[5] | **66.55** | **86.7** | **93.41** |
| BiPer | 61.53 | 83.14 | 93.75 |
| BORN[6] | **61.6** | **83.26** | **94.00** |
| VISPA | 62.1 | 83.40 | 92.8 |
| BORN[7] | **62.2** | **83.51** | **93.1** |

Table 2: GPT-2 benchmarks.

| Method | IMDB | WikiText-2 |
|---|---|---|
| | Acc.(%) | PPL |
| FP | 0.886 | 169.928 |
| IR-Net | 0.757 | 915.623 |
| STE | **0.831** | 319.990 |
| **BORN**[1] (ours) | 0.817 | **297.473** |
| LAB | 0.858 | 380.318 |
| **BORN**[2] (ours) | **0.878** | **273.473** |
| ReActNet | 0.869 | 275.870 |
| **BORN**[4] (ours) | **0.878** | **273.689** |

Table 4: Image Super-Resolution (x4): EDSR / SRResNet

| Method | Set5 | | Set14 | |
|---|---|---|---|---|
| | PSNR | SSIM | PSNR | SSIM |
| FP | 31.421 / 31.940 | 0.895 / 0.893 | 27.819 / 28.040 | 0.789 / 0.779 |
| IR-Net | 21.761 / 30.143 | 0.588 / 0.857 | 21.104 / 27.240 | 0.519 / 0.747 |
| STE | 28.460 / 29.760 | 0.810 / 0.860 | 26.030 / 27.060 | 0.710 / 0.750 |
| **BORN**[1] | **30.090** / **31.032** | **0.867** / **0.878** | **27.005** / **27.868** | **0.766** / **0.765** |
| LAB | 28.738 / **31.450** | 0.833 / 0.885 | 26.050 / **28.119** | 0.737 / 0.771 |
| **BORN**[2] | **29.403** / 31.428 | **0.848** / **0.895** | **26.524** / 27.834 | **0.748** / **0.789** |
| BBCU | 30.906 / **31.508** | 0.886 / 0.887 | 27.527 / 28.156 | 0.781 / 0.772 |
| **BORN**[3] | **30.983** / 31.505 | **0.887** / 0.887 | **27.597** / **28.160** | **0.782** / 0.772 |
| ReActNet | 29.302 / 29.296 | 0.840 / 0.845 | 26.477 / 26.775 | 0.741 / 0.738 |
| **BORN**[4] | **29.808** / **29.500** | **0.852** / **0.852** | **26.851** / **26.904** | **0.752** / **0.744** |

Superscripts in Table 2, Table 3, Table 4 denote the modification of base network topology according to a baseline algorithm, and using hyperparameters of baselines: [1]IR-Net (STE), [2]LAB, [3]BBCU, [4]ReActNet, [5]AdaBin, [6]BiPer, [7]VISPA.

BORN sustains high memory efficiency, with an average $\mu_m$ of 77.89%, ensuring consistent compression across architectures: 83.44% for EDSR, 71.5% for SRResNet, 64.46% for GPT-2, 92.16% for ResNet-18. The highest efficiency is observed across all ResNet-18-based architectures (92.16%), balancing quality and memory consumption. In contrast, GPT-2 architectures exhibit lower memory savings, suggesting that further optimization is needed for binarization in deep generative models.

BORN demonstrates a mean $\mu_p$ improvement of 17.05%, with per-architecture averages of 14.99% for EDSR, 9.45% for SRResNet, 38.1% for GPT-2, 5.67% for ResNet-18, ensuring faster convergence. The best stability is achieved by GPT-2 (IR-Net) (57.0%), showing strong optimization behavior in large-scale text modeling. However, ResNet-18 (BiPer) (2.86%) and GPT-2 (STE) (-10.5%) show weaker plateau behavior, indicating sensitivity to oscillations in their training dynamics.

Obtained results enable the application of BORN to arbitrary architectures, yielding robust and effective performance. Moreover, they demonstrate that employing a hybrid strategy—combining adaptive,static approximation and adaptive learning rate—further enhances stability and efficiency.

## 6 CONCLUSION

We proposed BORN, a training approach for BNNs that combines oscillations-aware gradient approximation with a binarization-specific adaptive LR. Without architectural modifications, it improves training stability and quality across diverse tasks. Experimental results demonstrate consistent improvement in standard quality metrics. For SR tasks, average PSNR improved by $4.81\%$ and SSIM by $6.52\%$ over SoTA BNN baselines. For language modeling, accuracy increased by up to $2.9\%$ on IMDB, and perplexity decreased by over $32.13\%$ on WikiText-2. For classification task, accuracy increased by up to $0.3\%$ on CIFAR-10 and $0.1\%$ on Imagenet. Theoretical analysis of BORN provides convergence guarantees and explains the superior descent behavior compared to STE.

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

## A  APPENDIX

## B  THEORETICAL RESULTS

### B.1  PROOF OF PROPOSITION 1

*Proof.* This result follows from (7) and basic calculations:

$$
\begin{aligned}
\nabla_{\bar{\alpha}} L_b(\boldsymbol{w})^T &= \left( \partial_i L\left(\operatorname{sign}(w_i)\right) g'_{\bar{\alpha}}(w_i) \right)_{i=1}^d \approx \\
&\approx \left( \partial_i L\left(g_{\bar{\alpha}}(w_i)\right) g'_{\bar{\alpha}}(w_i) \right)_{i=1}^d = \\
&= \nabla_{\boldsymbol{w}} L_{\bar{\alpha}}\left(\boldsymbol{w}\right)^T .
\end{aligned}
$$

$\square$

### B.2  PROOF OF THEOREM 1

*Proof.* The change of $\alpha_k$ implies that the approximation $g_{\alpha_k}$ is getting closer to $\operatorname{sign}$ as $k$ increases as in BORN. Considered algorithm during the phase $k$ with an approximation $g_{\alpha_k}$ for training a BNN works the following way: it yields a stationary point $\boldsymbol{w}^*_{\alpha_k}$ of (9). Since $\boldsymbol{w}^*_{\alpha_k}$ and $\boldsymbol{w}^*_{\bar{\alpha}}$ are the solutions of (9) with corresponding $\alpha$ values, then by the continuous dependence on the parameter, we get:

$$
\boldsymbol{w}^*_{\alpha_k} \xrightarrow[k \to +\infty]{} \boldsymbol{w}^*_{\bar{\alpha}} .
$$

$\square$

### B.3 PROOF OF COROLLARY 1

*Proof.* The approximation of sign function employed by STE corresponds to $g_{\alpha_0}$ with $\alpha_0 \leq \bar{\alpha}$. The training using STE yields a stationary point $\boldsymbol{w}^*_{\alpha_0}$ of (9) with parameter $\alpha = \alpha_0$. During the training of BORN, the sequence $\boldsymbol{w}^*_{\alpha_k}$ with $\alpha_k \xrightarrow[k \to +\infty]{} \bar{\alpha}$ is generated. By Theorem 1, this sequence converges to the optimal binary solution $\boldsymbol{w}^*_{\bar{\alpha}}$. Then, by the definition of minimum:

$$L_b\left(\boldsymbol{w}^*_{\bar{\alpha}}\right) \leq L_b\left(\boldsymbol{w}^*_{\alpha_0}\right),$$

which shows the superiority of BORN over STE. $\qquad \square$

### B.4 NOTATION AND AUXILIARY RESULTS FOR BORN CONVERGENCE RATE ANALYSIS

For the analysis of the convergence rate of BORN, we will use the following notation: $w_{ij}^{(k)}$ – weight for the connection from the $i$-th neuron in the $k$-th layer to the $j$-th neuron in the $(k+1)$-th layer, $a_i^{(k)}$ – weighted sum of the $i$-th neuron in the $k$-th layer. Since the network is two-layer, then $a_i^{(2)} = \hat{y}_i$, $\hat{\boldsymbol{y}} = (\hat{y}_1, \ldots, \hat{y}_m)$.

During the forward pass BORN exploits weights and activations binarization using:

$$Q_w(\boldsymbol{w}) = \frac{\|\boldsymbol{w}\|_1}{l(n+m)} \operatorname{sign}(\boldsymbol{w}), \ Q_a\left(a_i^{(k)}\right) = \operatorname{sign}\left(a_i^{(k)}\right).$$

Therefore,

$$a_i^{(1)} = \sum_{j=1}^n Q_w\left(w_{ji}^{(1)}\right) x_j, \ i = 1, \ldots, l,$$

$$a_i^{(2)} = \sum_{j=1}^l Q_w\left(w_{ji}^{(2)}\right) z_j, \ i = 1, \ldots, m,$$

where $z_j = Q_a\left(f(a_j^{(1)})\right)$, $\boldsymbol{x} = (x_1, \ldots, x_n)$.

During the backward pass BORN uses $g, z$ – the smooth approximations of sign function ($g$ for $Q_w$, $z$ for $Q_a$) according to (1), (2), (3), (4). In (4) let $F = 2.5, b = 1$.

**Lemma 3.** *The elements of $\nabla F$ equal to*

$$\frac{\partial F}{\partial w_{ij}^{(2)}} = \frac{2}{l(n+m)}\left(g'\left(w_{ij}^{(2)}\right)\|\boldsymbol{w}\|_1 + 1\right)(\hat{y}_j - y_j) z_i,$$

$$\frac{\partial F}{\partial w_{ij}^{(1)}} = \sum_{p=1}^m \frac{2}{l(n+m)} f'\left(a_j^{(1)}\right)\left(g'\left(w_{ij}^{(1)}\right)\|\boldsymbol{w}\|_1 + 1\right) \cdot$$

$$\cdot h'\left(f\left(a_j^{(1)}\right)\right) Q_w\left(w_{jp}^{(2)}\right) x_i (\hat{y}_p - y_p).$$

*Proof.* Everything is ready to compute $\nabla F$ with respect to weights:

$$\frac{\partial F}{\partial w_{ij}^{(2)}} = \frac{\partial F}{\partial \hat{y}_j} \frac{\partial \hat{y}_j}{\partial Q_w(w_{ij}^{(2)})} \frac{\partial Q_w\left(w_{ij}^{(2)}\right)}{\partial w_{ij}^{(2)}},$$

$$\frac{\partial F}{\partial \hat{y}_j} = 2\left(\hat{y}_j - y_j\right),$$

$$\frac{\partial \hat{y}_j}{\partial Q_w\left(w_{ij}^{(2)}\right)} = Q_a\left(f\left(a_i^{(1)}\right)\right) = z_i,$$

$$\frac{\partial Q_w\left(w_{ij}^{(2)}\right)}{\partial w_{ij}^{(2)}} = g'\left(w_{ij}^{(2)}\right) \frac{\|\boldsymbol{w}\|_1}{l(n+m)} + \frac{1}{l(n+m)}.$$

Thus, we have:

$$\frac{\partial F}{\partial w_{ij}^{(2)}} = \frac{2}{l(n+m)} \left( g'\left( w_{ij}^{(2)} \right) \|\boldsymbol{w}\|_1 + 1 \right) (\hat{y}_j - y_j) z_i. \tag{10}$$

Then, similarly:

$$\frac{\partial F}{\partial w_{ij}^{(1)}} = \sum_{p=1}^{m} \frac{\partial F}{\partial \hat{y}_p} \frac{\partial \hat{y}_p}{\partial z_j} \frac{\partial z_j}{\partial f\left(a_j^{(1)}\right)} f'\left(a_j^{(1)}\right) \frac{\partial Q_w\left(w_{ij}^{(1)}\right)}{\partial w_{ij}^{(1)}} x_i,$$

$$\frac{\partial F}{\partial w_{ij}^{(1)}} = \sum_{p=1}^{m} \frac{2}{l(n+m)} f'\left(a_j^{(1)}\right) \left( g'\left( w_{ij}^{(1)} \right) \|\boldsymbol{w}\|_1 + 1 \right) \cdot$$

$$\cdot \frac{\partial Q_a\left( f\left(a_j^{(1)}\right) \right)}{\partial f\left(a_j^{(1)}\right)} Q_w\left( w_{jp}^{(2)} \right) x_i (\hat{y}_p - y_p),$$

$$\frac{\partial F}{\partial w_{ij}^{(1)}} = \sum_{p=1}^{m} \frac{2}{l(n+m)} f'\left(a_j^{(1)}\right) \left( g'\left( w_{ij}^{(1)} \right) \|\boldsymbol{w}\|_1 + 1 \right) \cdot$$

$$\cdot h'\left( f\left(a_j^{(1)}\right) \right) Q_w\left( w_{jp}^{(2)} \right) x_i (\hat{y}_p - y_p). \tag{11}$$

$\square$

### B.5 PROOF OF LEMMA 1

*Proof.* Using Lemma 3, (10), (11), we compute $\|\nabla F\|^2$ :

$$\|\nabla F\|^2 = \frac{4}{l^2(n+m)^2} \sum_{i=1}^{l} \sum_{j=1}^{m} \left( g'\left( w_{ij}^{(2)} \right) \|\boldsymbol{w}\|_1 + 1 \right)^2 (\hat{y}_j - y_j)^2 z_i^2 +$$

$$+ \frac{4}{l^2(n+m)^2} \sum_{i=1}^{n} \sum_{j=1}^{l} f'^2\left(a_j^{(1)}\right) \left( g'\left( w_{ij}^{(1)} \right) \|\boldsymbol{w}\|_1 + 1 \right)^2 h'^2\left( f\left(a_j^{(1)}\right) \right) \cdot$$

$$\cdot x_i^2 \left( \sum_{p=1}^{m} Q_w\left( w_{jp}^{(2)} \right) (\hat{y}_p - y_p) \right)^2.$$

Since the following holds:

$$x_i^2 \leq 1, z_i^2 = 1,$$
$$f'(x) \leq 1,$$
$$h'(x) \leq 2.5,$$
$$\|\boldsymbol{w}\|_1 \leq \Omega l(n+m),$$
$$t_{\text{osc}} \geq 0,$$
$$g'(x) \leq t \leq T_{\max}^{\lambda_1 + \lambda_2} T_{\min}^{1 - \lambda_1 - \lambda_2} = 10^{2(\lambda_1 + \lambda_2) - 1},$$
$$\left( \sum_{p=1}^{m} Q_w\left( w_{jp}^{(2)} \right) (\hat{y}_p - y_p) \right)^2 \leq \left( \sum_{p=1}^{m} Q_w^2\left( w_{jp}^{(2)} \right) \right) \left( \sum_{p=1}^{m} (\hat{y}_p - y_p)^2 \right) \leq (m\Omega H)^2,$$

where the properties of OSA and SSA for upper bounds of $h'$ and $g'$ was used, then

$$\|\nabla F\|^2 \leq \frac{4mH^2 \left( 10^{2(\lambda_1 + \lambda_2) - 1} \Omega l(n+m) + 1 \right)^2}{l(n+m)^2} \left( 1 + 2.5^2 \Omega^2 mn \right).$$

$\square$

### B.6 PROOF OF LEMMA 2

*Proof.* By the definition of quantization error:

$$r_i = \frac{\|\boldsymbol{w}\|_1}{l(n+m)} \operatorname{sign}(w_i) - w_i,$$

$$r_i^2 = \frac{\|\boldsymbol{w}\|_1^2}{l^2(n+m)^2} + w_i^2 - 2\frac{\|\boldsymbol{w}\|_1}{l(n+m)} \operatorname{sign}(w_i)w_i \leq \frac{\|\boldsymbol{w}\|_1^2}{l^2(n+m)^2} + w_i^2 \leq 2\Omega^2,$$

$$\|\mathbf{r}\|^2 = \sum_{i=1}^{l(n+m)} r_i^2 \leq 2l(m+n)\Omega^2 = R^2.$$

$\square$

### B.7 PROOF OF THEOREM 2

*Proof.* We proof this theorem following to (Li et al., 2017). We start with estimating the upper bound of $\mathbb{E}\|\boldsymbol{w}^{\tau+1} - \boldsymbol{w}^*\|^2$. Let $\boldsymbol{w}_b^\tau = Q(\boldsymbol{w}^\tau)$. Taking expectation conditioned on $\boldsymbol{w}^\tau$, using convexity, Lipschitz condition and Cauchy-Schwarz inequality and upper bound for the quantization error, we get

$$\mathbb{E}\|\boldsymbol{w}^{\tau+1} - \boldsymbol{w}^*\|^2 = \mathbb{E}\|\boldsymbol{w}^\tau - \gamma_\tau\nabla F(\boldsymbol{w}_b^\tau) - \boldsymbol{w}^*\|^2 = \mathbb{E}\|\boldsymbol{w}^\tau - \gamma_\tau\nabla F(\boldsymbol{w}^\tau + \mathbf{r}^\tau) - \boldsymbol{w}^*\|^2 =$$

$$= \mathbb{E}\|\boldsymbol{w}^\tau - \gamma_\tau\nabla F(\boldsymbol{w}^\tau) + \gamma_\tau\nabla F(\boldsymbol{w}^\tau) - \gamma_\tau\nabla F(\boldsymbol{w}^\tau + \mathbf{r}^\tau) - \boldsymbol{w}^*\|^2 =$$

$$= \|\boldsymbol{w}^\tau - \boldsymbol{w}^*\|^2 - 2\gamma_\tau\mathbb{E}\langle\boldsymbol{w}^\tau - \boldsymbol{w}^*, \nabla F(\boldsymbol{w}^\tau)\rangle + 2\gamma_\tau\mathbb{E}\langle\boldsymbol{w}^\tau - \boldsymbol{w}^*, \nabla F(\boldsymbol{w}^\tau) -$$

$$- \nabla F(\boldsymbol{w}^\tau + \mathbf{r}^\tau)\rangle + \gamma_\tau^2\mathbb{E}\|\nabla F(\boldsymbol{w}^\tau + \mathbf{r}^\tau)\|^2 =$$

$$= \|\boldsymbol{w}^\tau - \boldsymbol{w}^*\|^2 - 2\gamma_\tau\langle\boldsymbol{w}^\tau - \boldsymbol{w}^*, \nabla L(\boldsymbol{w}^\tau)\rangle + 2\gamma_\tau\langle\boldsymbol{w}^\tau - \boldsymbol{w}^*, \nabla L(\boldsymbol{w}^\tau) -$$

$$- \nabla L(\boldsymbol{w}^\tau + \mathbf{r}^\tau)\rangle + \gamma_\tau^2\mathbb{E}\|\nabla F(\boldsymbol{w}^\tau + \mathbf{r}^\tau)\|^2 \leq$$

$$\leq \|\boldsymbol{w}^\tau - \boldsymbol{w}^*\|^2 - 2\gamma_\tau\langle\boldsymbol{w}^\tau - \boldsymbol{w}^*, \nabla L(\boldsymbol{w}^\tau)\rangle +$$

$$+ 2\gamma_\tau\|\boldsymbol{w}^\tau - \boldsymbol{w}^*\|\|\nabla L(\boldsymbol{w}^\tau) - \nabla L(\boldsymbol{w}^\tau + \mathbf{r}^\tau)\| +$$

$$+ \gamma_\tau^2 G^2 \leq \|\boldsymbol{w}^\tau - \boldsymbol{w}^*\|^2 - 2\gamma_\tau\langle\boldsymbol{w}^\tau - \boldsymbol{w}^*, \nabla L(\boldsymbol{w}^\tau)\rangle + 2\gamma_\tau R^2\mathcal{L}\|\boldsymbol{w}^\tau - \boldsymbol{w}^*\| + \gamma_\tau^2 G^2 \leq$$

$$\leq \|\boldsymbol{w}^\tau - \boldsymbol{w}^*\|^2 - 2\gamma_\tau(L(\boldsymbol{w}^\tau) - L(\boldsymbol{w}^*)) + 2\gamma_\tau R^2\mathcal{L}\|\boldsymbol{w}^\tau - \boldsymbol{w}^*\| + \gamma_\tau^2 G^2,$$

where in the second row we used:

$$-2\gamma_\tau^2\langle\nabla F(\boldsymbol{w}^\tau), \nabla F(\boldsymbol{w}^\tau) - \nabla F(\boldsymbol{w}^\tau + \mathbf{r}^\tau)\rangle + \gamma_\tau^2\|\nabla F(\boldsymbol{w}^\tau) - \nabla F(\boldsymbol{w}^\tau + \mathbf{r}^\tau)\|^2 +$$

$$+ \gamma_\tau^2\|\nabla F(\boldsymbol{w}^\tau)\|^2 = \gamma_\tau^2\|\nabla F(\boldsymbol{w}^\tau + \mathbf{r}^\tau)\|^2.$$

Rearranging the terms, we get:

$$L(\boldsymbol{w}^\tau) - L(\boldsymbol{w}^*) \leq \frac{1}{2\gamma_\tau}\left(\|\boldsymbol{w}^\tau - \boldsymbol{w}^*\|^2 + \mathcal{L}R^2\|\boldsymbol{w}^\tau - \boldsymbol{w}^*\|^2 + \gamma_\tau^2 G^2 - \mathbb{E}\|\boldsymbol{w}^{\tau+1} - \boldsymbol{w}^*\|^2\right).$$

Taking the expectation:

$$\mathbb{E}(L(\boldsymbol{w}^\tau) - L(\boldsymbol{w}^*)) \leq$$

$$\leq \frac{1}{2\gamma_\tau}\left(\mathbb{E}\|\boldsymbol{w}^\tau - \boldsymbol{w}^*\|^2 + \mathcal{L}R^2\mathbb{E}\|\boldsymbol{w}^\tau - \boldsymbol{w}^*\|^2 + \gamma_\tau^2 G^2 - \mathbb{E}\|\boldsymbol{w}^{\tau+1} - \boldsymbol{w}^*\|^2\right).$$

Sum the inequality by $\tau = 1, \ldots, T$:

$$\sum_{\tau=1}^T \mathbb{E}(L(\boldsymbol{w}^\tau) - L(\boldsymbol{w}^*)) \leq \frac{1}{\gamma_1}\mathbb{E}\|\boldsymbol{w}^1 - \boldsymbol{w}^*\|^2 + \sum_{\tau=2}^T\left(\frac{1}{2\gamma_\tau} - \frac{1}{2\gamma_{\tau-1}}\right)\mathbb{E}\|\boldsymbol{w}^\tau - \boldsymbol{w}^*\|^2 +$$

$$+ \sum_{\tau=1}^T \frac{\gamma_\tau}{2}G^2 + \frac{\mathcal{L}R^2}{2}\sum_{\tau=1}^T \mathbb{E}\|\boldsymbol{w}^\tau - \boldsymbol{w}^*\|^2.$$

Since $\left| w_{ij}^{(k)} \right| \leq \Omega$, then $\| \boldsymbol{w}^\tau - \boldsymbol{w}^* \|^2 \leq 4l(m+n)\Omega^2$. Thus,

$$\sum_{\tau=1}^{T} \mathbb{E}(L(\boldsymbol{w}^\tau) - L(\boldsymbol{w}^*)) \leq \frac{4l(m+n)\Omega^2}{\gamma_1} + 4l(m+n)\Omega^2 \left( \frac{1}{2\gamma_T} - \frac{1}{2\gamma_1} \right) +$$

$$+ \frac{G^2}{2} \sum_{\tau=1}^{T} \gamma_\tau + 4l(m+n)T\Omega^2 \frac{\mathcal{L}R^2}{2}.$$

Let $\overline{\boldsymbol{w}}^T = \frac{1}{T} \sum_{\tau=1}^{T} \boldsymbol{w}^\tau$. Using Jensen's inequality:

$$\mathbb{E}(L(\overline{\boldsymbol{w}}^T) - L(\boldsymbol{w}^*)) \leq \frac{1}{T} \sum_{\tau=1}^{T} \mathbb{E}(L(\boldsymbol{w}^\tau) - L(\boldsymbol{w}^*)),$$

we get:

$$\mathbb{E}(L(\overline{\boldsymbol{w}}^T) - L(\boldsymbol{w}^*)) \leq \frac{1}{T} \sum_{\tau=1}^{T} (L(\boldsymbol{w}^\tau) - L(\boldsymbol{w}^*)) \leq$$

$$\leq \frac{4l(m+n)\Omega^2}{T\gamma_1} + \frac{4l(m+n)\Omega^2}{T} \left( \frac{1}{2\gamma_T} - \frac{1}{2\gamma_1} \right) + \frac{G^2}{2T} \sum_{\tau=1}^{T} \gamma_\tau + 4l(m+n)\Omega^2 \frac{\mathcal{L}R^2}{2}.$$

$\square$

### B.8 Convergence rate estimate of Straight-Through Estimator

As an additional result we provide the convergence rate estimate for STE. In order to do that, consider the same neural network architecture, assumptions on $\boldsymbol{x}_i, \boldsymbol{y}_i, \hat{\boldsymbol{y}}_i$ and notation used while studying BORN: $w_{ij}^{(k)}$ – weight for the connection from the $i$-th neuron in the $k$-th layer to the $j$-th neuron in the $(k+1)$-th layer, $a_i^{(k)}$ – weighted sum of the $i$-th neuron in the $k$-th layer, $\hat{\boldsymbol{y}} = (\hat{y}_1, \ldots, \hat{y}_m)$, $\boldsymbol{x} = (x_1, \ldots, x_n)$. Since the network is two-layer, then $a_i^{(2)} = \hat{y}_i$. However, now $z_j = \text{sign}(f(a_j^{(1)}))$. During the forward pass weight binarization is applied using

$$Q(\boldsymbol{w}) = \frac{\| \boldsymbol{w} \|_1}{l(n+m)} \text{sign}(\boldsymbol{w}),$$

and all the weighted sums are computed using binary weights $Q(w_{ij}^{(k)})$. Therefore, we will assume that

$$a_i^{(1)} = \sum_{j=1}^{n} Q\left( w_{ji}^{(1)} \right) x_j, \ i = 1, \ldots, l,$$

$$a_i^{(2)} = \sum_{j=1}^{l} Q\left( w_{ji}^{(2)} \right) z_j, \ i = 1, \ldots, m.$$

During the backward pass it is necessary to use the derivative of sign. STE implies identity function as the sign approximation:

$$g(x) = x \cdot \mathbf{1}_{[-1;1]}(x), \ g'(x) = \mathbf{1}_{[-1;1]}(x),$$

where $\mathbf{1}_{[-1;1]}$ is the indicator function of $[-1; 1] \subset \mathbb{R}$.

**Lemma 4.** *The elements of $\nabla F$ equal to*

$$\frac{\partial F}{\partial w_{ij}^{(2)}} = \frac{2}{l(n+m)} \left( \mathbf{1}_{[-1;1]} \left( w_{ij}^{(2)} \right) \| \boldsymbol{w} \|_1 + 1 \right) (\hat{y}_j - y_j) z_i,$$

$$\frac{\partial F}{\partial w_{ij}^{(1)}} = \sum_{p=1}^{m} \frac{2\| \boldsymbol{w} \|_1}{l^2(n+m)^2} f'\left( a_j^{(1)} \right) \left( \mathbf{1}_{[-1;1]} \left( w_{ij}^{(1)} \right) \| \boldsymbol{w} \|_1 + 1 \right) \cdot$$

$$\cdot \mathbf{1}_{[-1;1]} \left( f\left( a_j^{(1)} \right) \right) \text{sign} \left( w_{jp}^{(2)} \right) x_i (\hat{y}_p - y_p).$$

*Proof.* Everything is ready to compute $\nabla F$ with respect to weights.

$$\frac{\partial F}{\partial w_{ij}^{(2)}} = \frac{\partial F}{\partial \hat{y}_j} \cdot \frac{\partial \hat{y}_j}{\partial Q\left(w_{ij}^{(2)}\right)} \cdot \frac{\partial Q\left(w_{ij}^{(2)}\right)}{\partial w_{ij}^{(2)}},$$

$$\frac{\partial F}{\partial \hat{y}_j} = 2\left(\hat{y}_j - y_j\right),$$

$$\frac{\partial \hat{y}_j}{\partial Q\left(w_{ij}^{(2)}\right)} = z_i,$$

$$\frac{\partial Q\left(w_{ij}^{(2)}\right)}{\partial w_{ij}^{(2)}} = \mathbf{1}_{[-1;1]}(w_{ij}^{(2)}) \cdot \frac{\|\boldsymbol{w}\|_1}{l(n+m)} + \frac{1}{l(n+m)}.$$

Thus, we have

$$\frac{\partial F}{\partial w_{ij}^{(2)}} = \frac{2}{l(n+m)} \left(\mathbf{1}_{[-1;1]}\left(w_{ij}^{(2)}\right)\|\boldsymbol{w}\|_1 + 1\right)\left(\hat{y}_j - y_j\right) z_i. \tag{12}$$

Then, similarly

$$\frac{\partial F}{\partial w_{ij}^{(1)}} = \sum_{p=1}^{m} \frac{\partial F}{\partial \hat{y}_p} \frac{\partial \hat{y}_p}{\partial z_j} \frac{\partial z_j}{\partial f\left(a_j^{(1)}\right)} f'\left(a_j^{(1)}\right) \frac{\partial Q\left(w_{ij}^{(1)}\right)}{\partial w_{ij}^{(1)}} x_i,$$

$$\frac{\partial F}{\partial w_{ij}^{(1)}} = \sum_{p=1}^{m} \frac{2}{l(n+m)} f'\left(a_j^{(1)}\right)\left(\mathbf{1}_{[-1;1]}\left(w_{ij}^{(1)}\right)\|\boldsymbol{w}\|_1 + 1\right) \cdot$$

$$\cdot\mathbf{1}_{[-1;1]}\left(f\left(a_j^{(1)}\right)\right) Q\left(w_{jp}^{(2)}\right) x_i \left(\hat{y}_p - y_p\right),$$

$$\frac{\partial F}{\partial w_{ij}^{(1)}} = \sum_{p=1}^{m} \frac{2\|\boldsymbol{w}\|_1}{l^2(n+m)^2} f'\left(a_j^{(1)}\right)\left(\mathbf{1}_{[-1;1]}\left(w_{ij}^{(1)}\right)\|\boldsymbol{w}\|_1 + 1\right) \cdot$$

$$\cdot\mathbf{1}_{[-1;1]}\left(f\left(a_j^{(1)}\right)\right) \operatorname{sign}\left(w_{jp}^{(2)}\right) x_i(\hat{y}_p - y_p). \tag{13}$$

$\square$

Suppose that weights domain has a finite diameter: there exists a constant $\Omega$, such that for all $i$, $j$, $k$ inequality $|w_{ij}^{(k)}| \leq \Omega$ holds. Consequently, for all $i$ $|\hat{y}_i|$ are also bounded.

**Lemma 5.** *If weights binarization is applied according to STE (see Appendix B for more details), then*

$$\|\nabla F\|^2 \leq \frac{4mH^2\left(1 + \Omega l(n+m)\right)^2}{l(n+m)^2}\left(\Omega^2 mn + 1\right) = \hat{G}^2,$$

*where* $|w_{ij}^{(k)}| \leq \Omega, (\hat{y}_p - y_p)^2 \leq H^2$.

*Proof.* Using Lemma 4, (12), (13), we compute $\|\nabla F\|^2$.

$$\|\nabla F\|^2 = \frac{4}{l^2(n+m)^2} \sum_{i=1}^{l}\sum_{j=1}^{m}\left(\mathbf{1}_{[-1;1]}\left(w_{ij}^{(2)}\right)\|\boldsymbol{w}\|_1 + 1\right)^2 (\hat{y}_j - y_j)^2 z_i^2 +$$

$$+ \frac{4\|\boldsymbol{w}\|_1^2}{l^4(n+m)^4} \sum_{i=1}^{n}\sum_{j=1}^{l} f'^2\left(a_j^{(1)}\right)\left(\mathbf{1}_{[-1;1]}\left(w_{ij}^{(1)}\right)\|\boldsymbol{w}\|_1 + 1\right)^2 \cdot$$

$$\cdot\mathbf{1}_{[-1;1]}\left(f\left(a_j^{(1)}\right)\right) x_i^2 \left(\sum_{p=1}^{m} \operatorname{sign}\left(w_{jp}^{(2)}\right)(\hat{y}_p - y_p)\right)^2.$$

Since

$$x_i^2 \le 1, z_i^2 = 1,$$
$$f'(x) \le 1, \mathbf{1}_{[-1;1]}(x) \le 1,$$
$$\|\boldsymbol{w}\|_1 \le \Omega l(n+m),$$
$$\mathbf{1}_{[-1;1]}(w_{ij}^{(k)})\|\boldsymbol{w}\|_1 + 1 \le \|\boldsymbol{w}\|_1 + 1 \le 1 + \Omega l(n+m),$$
$$\left(\sum_{p=1}^{m} \operatorname{sign}\left(w_{jp}^{(2)}\right)(\hat{y}_p - y_p)\right)^2 \le \left(\sum_{p=1}^{m} \operatorname{sign}^2\left(w_{jp}^{(2)}\right)\right)\left(\sum_{p=1}^{m}(\hat{y}_p - y_p)^2\right) \le (mH)^2,$$

then

$$\|\nabla F\|^2 \le \frac{4}{l^2(n+m)^2}\left(lm\left(1 + \Omega l(n+m)\right)^2 H^2 + \Omega^2 ln\left(1 + \Omega l(n+m)\right)^2 (mH)^2\right) =$$
$$= \frac{4mH^2\left(1 + \Omega l(n+m)\right)^2}{l(n+m)^2}\left(\Omega^2 mn + 1\right)$$
$$\|\nabla F\|^2 \le \frac{4mH^2\left(1 + \Omega l(n+m)\right)^2}{l(n+m)^2}\left(\Omega^2 mn + 1\right) = \hat{G}^2.$$

$\square$

**Lemma 6.** *The following inequality for the quantization error norm holds:*
$$\|\mathbf{r}\|^2 \le 2l(m+n)\Omega^2 = \hat{R}^2.$$

*Proof.* By the definition of quantization error

$$r_i = \frac{\|\boldsymbol{w}\|_1}{l(n+m)}\operatorname{sign}(w_i) - w_i,$$
$$r_i^2 = \frac{\|\boldsymbol{w}\|_1^2}{l^2(n+m)^2} + w_i^2 - 2\frac{\|\boldsymbol{w}\|_1}{l(n+m)}\operatorname{sign}(w_i)w_i \le \frac{\|\boldsymbol{w}\|_1^2}{l^2(n+m)^2} + w_i^2 \le 2\Omega^2,$$
$$\|\mathbf{r}\|^2 = \sum_{i=1}^{l(n+m)} r_i^2 \le 2l(m+n)\Omega^2 = \hat{R}^2.$$

$\square$

**Theorem 3.** *Let $\{\boldsymbol{w}^\tau\}$ be the sequence of weights generated by SGD with STE and learning rates $\{\gamma_\tau\}$ such that $\boldsymbol{w}^\tau \xrightarrow[\tau \to \infty]{} \boldsymbol{w}^*$. Assume the loss $L$ is convex and its gradient is Lipschitz continuous with constant $\mathcal{L}$. Then:*

$$\mathbb{E}(L(\overline{\boldsymbol{w}}^T) - L(\boldsymbol{w}^*)) \le \frac{4l(m+n)\Omega^2}{T\gamma_1} + \frac{4l(m+n)\Omega^2}{T}\left(\frac{1}{2\gamma_T} - \frac{1}{2\gamma_1}\right)$$
$$+ \frac{\hat{G}}{2T}\sum_{\tau=1}^{T}\gamma_\tau + 2l(m+n)\Omega^2 \mathcal{L}\hat{R}^2,$$

*where $\overline{\boldsymbol{w}}^T = \frac{1}{T}\sum_{\tau=1}^{T}\boldsymbol{w}^\tau$.*

*Proof.* Proof of this theorem is the same to the proof of Theorem 2. $\square$

### B.9 CONVERGENCE RATE ESTIMATE OF IR-NET

Convergence rate estimate can be obtained for IR-Net similarly to BORN and STE.

During forward pass IR-Net exploits the following weight binarization function:

$$Q(\mathbf{w}) = \operatorname{sign}(\boldsymbol{w}) \cdot 2^{\left\lfloor \log_2\left(\frac{\|\boldsymbol{w}\|_1}{nl+lm}\right)\right\rceil}.$$

During the backward pass IR-Net uses $g$ – the differentiable approximation of sign:

$$g(x) = k \tanh(tx), \; g'(x) = kt(1 - \tanh^2(tx)),$$

where $t = T_{\min} \cdot 10^{\frac{i}{K} \cdot \log \frac{T_{\max}}{T_{\min}}}$, $T_{\min} = 10^{-1}$, $T_{\max} = 10$, $k = \max(\frac{1}{t}, 1)$, $i$ – epoch number, $K$ – number of epochs.

**Lemma 7.** *If weights binarization is applied according to IR-Net, then*

$$\|\nabla F\|^2 \leq 160000 lm H^2 \Omega^2 (1 + 10000 mn) = \tilde{G}^2,$$

*where* $|w_{ij}^{(k)}| \leq \Omega$, $(\hat{y}_p - y_p)^2 \leq \tilde{H}^2$.

**Lemma 8.** *The following inequality for the quantization error norm holds:*

$$\|\mathbf{r}\|^2 \leq 2l(m + n)\Omega^2 = \hat{R}^2.$$

**Theorem 4.** *Let* $\{\boldsymbol{w}^\tau\}$ *be the sequence of weights generated by SGD with IR-Net and learning rates* $\{\gamma_\tau\}$ *such that* $\boldsymbol{w}^\tau \xrightarrow[\tau \to \infty]{} \boldsymbol{w}^*$. *Assume the loss $L$ is convex and its gradient is Lipschitz continuous with constant $\mathcal{L}$. Then:*

$$\mathbb{E}(L(\overline{\boldsymbol{w}}^T) - L(\boldsymbol{w}^*)) \leq \frac{4l(m+n)\Omega^2}{T\gamma_1} + \frac{4l(m+n)\Omega^2}{T}\left(\frac{1}{2\gamma_T} - \frac{1}{2\gamma_1}\right)$$

$$+ \frac{\tilde{G}}{2T}\sum_{\tau=1}^{T}\gamma_\tau + 2l(m+n)\Omega^2 \mathcal{L}\tilde{R}^2,$$

*where* $\overline{\boldsymbol{w}}^T = \frac{1}{T}\sum_{\tau=1}^{T}\boldsymbol{w}^\tau$.

### B.10 BORN AND STRAIGHT-THROUGH ESTIMATOR CONVERGENCE RATE ESTIMATES COMPARISON

Convergence rate estimates of BORN and STE were derived. Main theorem explains that the training of BNN converges until it reaches the accuracy floor of quantization error bound $l(m+n)\Omega^2 \mathcal{L}R^2$, which is unavoidable because of the quantization.

The next step is to compare the estimates in order to find out which binarization model performs better in terms of convergence rate.

**Lemma 9.** *For every $T \in \mathbb{N}$ the following inequality holds*

$$\sum_{\tau=1}^{T}\frac{1}{\sqrt{\tau}} \leq 2\sqrt{T} - 1.$$

*Proof.* We will use mathematical induction. The base case $T = 1$ is trivial. Assume that the inequality holds for $T$ and consider $T + 1$.

$$\sum_{\tau=1}^{T+1}\frac{1}{\sqrt{\tau}} = \sum_{\tau=1}^{T}\frac{1}{\sqrt{\tau}} + \frac{1}{\sqrt{T+1}} \leq 2\sqrt{T} - 1 + \frac{1}{\sqrt{T+1}},$$

$$2\sqrt{T} - 1 + \frac{1}{\sqrt{T+1}} \leq 2\sqrt{T+1} - 1,$$

$$2\sqrt{T} + \frac{1}{\sqrt{T+1}} \leq 2\sqrt{T+1},$$

$$2\sqrt{T(T+1)} + 1 \leq 2(T+1),$$

$$4T(T+1) \leq (2T+1)^2,$$

$$4T^2 + 4T \leq 4T^2 + 4T + 1,$$

$$0 \leq 1,$$

which completes the proof. $\square$

Proven lemma helps us to simplify the expression for the estimates:

$$\mathbb{E}(L(\overline{\boldsymbol{w}}^T) - L(\boldsymbol{w}^*)) \leq \frac{4l(m+n)\Omega^2}{T\gamma_1} + \frac{4l(m+n)\Omega^2}{T}\left(\frac{1}{2\gamma_T} - \frac{1}{2\gamma_1}\right) +$$

$$+ \frac{G^2}{2T}\sum_{\tau=1}^{T}\gamma_\tau + 4l(m+n)\Omega^2\frac{\mathcal{L}R^2}{2} \leq \frac{4l(m+n)\Omega^2}{cT} + \frac{4l(m+n)\Omega^2}{T}\left(\frac{\sqrt{T}}{2c} - \frac{1}{2c}\right) +$$

$$+ \frac{G^2}{2T}\cdot c(2\sqrt{T}-1) + 4l(m+n)\Omega^2\frac{\mathcal{L}R^2}{2}.$$

BORN convergence rate estimate:

$$\mathbb{E}(L(\overline{\boldsymbol{w}}^T) - L(\boldsymbol{w}^*)) \leq \frac{4l(m+n)\Omega^2}{cT} + \frac{2l(m+n)\Omega^2}{T}\left(\frac{\sqrt{T}}{c} - \frac{1}{c}\right) +$$

$$+ \frac{G_1^2}{2T}\cdot c(2\sqrt{T}-1) + 2l(m+n)\Omega^2\mathcal{L}R^2,$$

where

$$G_1^2 = \frac{4mH^2\left(10^{2(\lambda_1+\lambda_2)-1}\Omega l(n+m)+1\right)^2}{l(n+m)^2}\left(1 + 2.5^2\Omega^2 mn\right),$$

$$R^2 = 2l(m+n)\Omega^2.$$

STE convergence rate estimate:

$$\mathbb{E}(L(\overline{\boldsymbol{w}}^T) - L(\boldsymbol{w}^*)) \leq \frac{4l(m+n)\Omega^2}{cT} + \frac{2l(m+n)\Omega^2}{T}\left(\frac{\sqrt{T}}{c} - \frac{1}{c}\right) +$$

$$+ \frac{\hat{G}}{2T}\cdot c(2\sqrt{T}-1) + 2l(m+n)\Omega^2\mathcal{L}\hat{R}^2,$$

where

$$\hat{G} = \frac{4mH^2\left(1+\Omega l(n+m)\right)^2}{l(n+m)^2}\left(\Omega^2 mn + 1\right),$$

$$\hat{R}^2 = 2l(m+n)\Omega^2.$$

Assume that Lipschitz constants, $\Omega, l, m, n$ for both cases are the same. Then the comparison of convergence rate estimates comes down to comparison of $G^2$ and $\hat{G}^2$ – gradient bound constants for BORN and STE, correspondingly. However, it is obvious that $G^2 \geq \hat{G}^2$. Thus, the convergence rate estimate for STE is smaller than for BORN, but there is no contradiction with the results in Section 4.1 and (Yin et al., 2019), since the STE is more rough approximation for the considered BNN training task. This rough method can lead to a faster convergence, but to the worse weight vector in terms of loss function value comparing with BORN, which is a result of Corollary 1.

### B.11 CONVERGENCE RATE ESTIMATE IN PRACTICE

To reinforce the results of the theoretical results on the convergence rate, the same fully connected network was implemented to check if the estimate holds in practice. In particular, we set the number of hidden neurons $l$ to 16, $\Omega, H, m, n, \mathcal{L}$ to 1. For training data, we chose 50 points $(x_i, \sin(x_i))$, where $x_i$ are evenly spaced in $\left[0; \frac{\pi}{4}\right]$. For validation data, we choose 1000 points $(x_i, \cos(x_i))$, where $x_i$ are evenly spaced in $\left[0; \frac{\pi}{4}\right]$. We set batch size to 1, and train the network for 400 epochs with SGD and MSE loss. The optimal loss value $L(\boldsymbol{w}^*)$ was estimated as the minimal loss value during the training. At each iteration $T$, we calculated $L(\overline{\boldsymbol{w}}^T)$. Figure 4 demonstrates the validity of theoretical estimate.

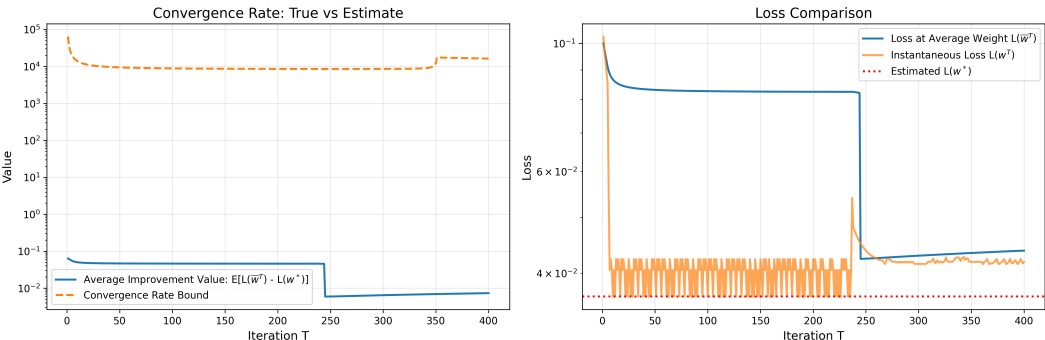

Figure 4: Comparison of true loss improvement value and estimated convergence rate. The plot on the right presents the loss at each iteration and the corresponding loss at the average weight. Log scale was used in both plots.

## C    RELATED WORK COMPARISON TABLES

Table 5: Accuracy comparison of SoTA methods on ImageNet dataset and ResNet18 architecture.

| Method | ImageNet-ResNet18 | |
|---|---|---|
| | Top-1 (%) | Top-5 (%) |
| FP | 69.6 | 89.2 |
| BNN (Courbariaux et al., 2016) | 42.2 | – |
| IR-Net (Qin et al., 2020) | 58.1 | 80.0 |
| BinaryDuo (Kim et al., 2020) | 60.4 | 82.3 |
| BiPer (Vargas et al., 2024) | 61.4 | 83.14 |
| ReBNN (Xu et al., 2023) | 61.6 | 83.4 |
| VISPA (Orecchia et al., 2024) | 62.1 | 83.4 |
| AdaBin (Tu et al., 2022) | 63.1 | 84.3 |
| LAB (Falkena et al., 2023) | 64.2 | 85.0 |
| ReActNet (Tu et al., 2022) | 65.5 | 86.1 |

Table 6: Performance comparison of SoTA methods for image SR task on Set5, Set14 datasets and EDSR, SRResNet architectures.

| Topology | Method | Set5 | | Set14 | |
|---|---|---|---|---|---|
| | | PSNR | SSIM | PSNR | SSIM |
| SRResNet (x4 scale) | FP (Ledig et al., 2017) | 32.16 | 0.8951 | 28.60 | 0.7822 |
| | BNN (Courbariaux et al., 2016) | 27.56 | 0.7896 | 25.51 | 0.6820 |
| | E2FIF (Song et al., 2023) | 31.33 | 0.880 | 27.93 | 0.766 |
| | IR-Net (Qin et al., 2020) | 31.38 | 0.8835 | 28.08 | 0.7679 |
| | ReActNet (Liu et al., 2020) | 31.54 | 0.8859 | 28.19 | 0.7705 |
| | BBCU (Xia et al., 2022) | **31.79** | **0.8905** | **28.38** | **0.7762** |
| EDSR (x4 scale) | FP (Lim et al., 2017) | 32.46 | 0.897 | 28.80 | 0.787 |
| | BNN (Courbariaux et al., 2016) | 17.53 | 0.188 | 17.51 | 0.160 |
| | E2FIF (Song et al., 2023) | **31.91** | **0.890** | **28.29** | **0.775** |

## D    EXPERIMENTAL HYPERPARAMETERS

To apply our method to SRResNet, EDSR and GPT-2, we incorporated architectural modifications of each baseline algorithm (BBCU, IR-Net, LAB, ReActNet, STE) while adjusting the training process and convolutional approximations. By STE we meant hardtanh (Courbariaux et al., 2016) sign approximation and scaling factor for weights (Rastegari et al., 2016). In the SR task, we followed the architectural adjustments of each method. IR-Net, LAB, and STE do not introduce any architectural changes, so no modifications were required. BBCU employs RSign, RPReLU, additional residual connections, and bilinear approximation in the final layer, while ReActNet utilizes RSign, RPReLU, and additional residual connections. Since LAB focuses solely on activation binarization, we applied

the ReActNet method for weight binarization, following the approach used in the LAB-BNN model described in (Falkena et al., 2023). For the GPT-2 language model, we adopted only the RSign mechanism from ReActNet. Additionally, we replaced Conv1d convolutions with adapted versions specific to each method, effectively transforming Conv2d into Conv1d for LLMs while preserving the core principles of each approach. For ResNet-18, binarization was performed in accordance with established baselines. As baselines for GPT-2, we considered only those methods that have historically been applied to two or more tasks (Xia et al., 2022).

In our work, for image SR we focus on 4x image quality enhancement. Each mini-batch consists of 16 images. In the BBCU, IR-Net, LAB, ReActNet, and STE, the image size of 128x128 was used in the SRResNet model, whereas in all other cases, a 192x192 image size was employed. For training, we utilize the $L_1$ loss as our primary metric to ensure accurate pixel-wise reconstruction and promote sharper image details. Appendix D, Table 7 lists the remaining hyperparameters. Unless specified, training settings followed PyTorch 2.4.1 defaults. For BBCU SRResNet, hyperparameters were taken from its original paper. For IR-Net, LAB, and ReActNet, originally designed for image recognition, we selected the training hyperparameters ourselves to optimize performance on SR and NLP tasks. ReActNet was trained in two stages, resetting the scheduler at each stage. IR-Net EDSR used 64-bit precision due to training instability. We restrict our quantitative evaluation to the Set5 and Set14 datasets, as these benchmarks are commonly adopted for numerical comparison in related works addressing similar tasks (Adapa et al., 2025; Jiang et al., 2024). For ResNet-18 on CIFAR-10 and ImageNet, we adopted the same hyperparameter settings as reported in the corresponding baseline works (see Table 9).

A default linear LR scheduler is applied across all experimental runs. Additionally, we consistently utilize the AdamW optimizer, configured with betas of (0.9, 0.999) and an epsilon value of 1e-8. Appendix D, Table 8 summarizes the hyperparameters used in our experiments. Unless specified otherwise, the remaining training parameters and configurations were adopted from the default settings provided by the corresponding modules in the Transformers library version 4.45.2. All runs were made on a single NVIDIA A100 with 40GB GPU memory.

Table 7: Hyperparameters for image super-resolution

| neural network | optimizer (Adam) | scheduler | epochs | lr range | hr range | $T_{\min}$ | $T_{\max}$ | $F$ | $b$ |
|---|---|---|---|---|---|---|---|---|---|
| SRResNet (ReAct) | betas=(0.9, 0.99), lr=1e-4 | StepLR (step_size=1000, gamma=0.5) | 6000 | imagenet norm | $[-1, 1]$ | 0.1 | 10 | 2.5 | 2 |
| SRResNet (LAB/IR-Net/STE) | betas=(0.9, 0.99), lr=1e-4 | StepLR (step_size=4000, gamma=0.5) | 6000 | imagenet norm | $[-1, 1]$ | 0.1 | 10 | 2.5 | 2.5 |
| SRResNet (BBCU) | betas=(0.9, 0.99), lr=1e-4 | StepLR (step_size=4000, gamma=0.5) | 6000 | $[0, 1]$ | $[0, 1]$ | 0.1 | $10^{1.5}$ | 2 | 1.5 |
| EDSR (IR-Net, STE) | betas=(0.9, 0.999), lr=1e-4 | StepLR (step_size=50, gamma=0.5) | 200 | $[0, 1]$ | $[0, 1]$ | 0.1 | 10 | 2.5 | 1 |
| EDSR (ReAct) | params=filter(lambda p: p.requires_grad), lr=1e-4 | StepLR (step_size=150, gamma=0.5) | 200 | $[0, 1]$ | $[0, 1]$ | 0.1 | 10 | 2.5 | 2 |
| EDSR (LAB) | betas=(0.9, 0.999), lr=1e-4 | StepLR (step_size=55, gamma=0.5) | 200 | $[0, 1]$ | $[0, 1]$ | 0.1 | 10 | 2.5 | 2.5 |
| EDSR (BBCU) | betas=(0.9, 0.999), lr=2e-4 | CosineAnnealingLR (T_max=200, eta_min=1e-7) | 200 | $[0, 1]$ | $[0, 1]$ | 0.1 | $10^{1.5}$ | 2.5 | 1 |

lr – learning rate, betas – Adam coefficients, step_size – LR decay interval, gamma – LR decay factor, T_max – cosine annealing period, eta_min – min LR. $T_{\min}$ and $T_{\max}$ are hyperparameters of OSA, $F$ and $b$ are hyperparameters of SSA.

Table 8: Hyperparameters for LLMs.

| Dataset | Baseline NN | Optimizer (AdamW) | Scheduler | Epochs | Batch | $T_{\min}$ | $T_{\max}$ | $F$ | $b$ |
|---|---|---|---|---|---|---|---|---|---|
| IMDB | GPT-2 (IR-Net/STE) | lr=5e-4 | LinearLR | 10 | 8 | 0.1 | 10 | 2.5 | 2.5 |
| | GPT-2 (ReActNet/ BBCU/LAB) | lr=5e-5 | LinearLR | 10 | 8 | 0.1 | 10 | 2.5 | 2.5 |
| WIKITEXT | GPT-2 (IR-Net/STE) | lr=5e-4 | LinearLR | 50 | 7 | 0.1 | 10 | 2.5 | 2.5 |
| | GPT-2 (ReActNet/ BBCU/LAB) | lr=5e-5 | LinearLR | 50 | 7 | 0.1 | 10 | 2.5 | 2.5 |

Table 9: Hyperparameters for CLF model.

| Dataset | Baseline NN | Optimizer | Scheduler | Epochs | Batch | $T_{\min}$ | $T_{\max}$ | $F$ | $b$ |
|---|---|---|---|---|---|---|---|---|---|
| CIFAR-10 | ResNet-18 (IR-Net/STE) | Adam betas=(0.9, 0.99), lr=1e-2 | CosineAnnealingLR (eta_min=1e-7) | 400 | 128 | 0.1 | 10 | 1.75 | 1 |
| Imagenet | ResNet-18 (IR-Net/STE) | SGD momentum=0.9, weight decay=1e-5, lr=0.2 | StepLR (step_size=18750, gamma=0.1) | 1 | 64 | 0.1 | 10 | 1.75 | 1 |
| CIFAR-10 | ResNet-18 (LAB) | Adam betas=(0.9, 0.99), lr=1e-3 | CosineAnnealingLR (eta_min=1e-7) | 200 | 128 | 0.1 | 10 | 1.75 | 1 |
| Imagenet | ResNet-18 (LAB) | Adam betas=(0.9, 0.999), lr=2.5e-3 | CosineAnnealingLR (eta_min=0) | 300 | 256 | 0.1 | 10 | 1.75 | 1 |
| CIFAR-10 | ResNet-18 (ReActNet) | Adam betas=(0.9, 0.999), lr=5e-4 | LambdaLR( lambda: 1-epoch/epochs) | 120 | 512 | 0.1 | 10 | 1.75 | 1 |
| Imagenet | ResNet-18 (ReActNet) | Adam betas=(0.9, 0.999), lr=5e-4 | LambdaLR( lambda: 1-epoch/epochs) | 120 | 512 | 0.1 | 10 | 1.75 | 1 |
| CIFAR-10 | ResNet-18 (AdaBin) | SGD momentum=0.9, weight decay=5e-4, lr=5e-4 | CosineAnnealingLR (eta_min=0) | 200 | 128 | 0.1 | 10 | 1.75 | 1 |
| Imagenet | ResNet-18 (AdaBin) | SGD momentum=0.9, weight decay=1e-4, lr=1e-1 | CosineAnnealingLR (eta_min=0) | 120 | 128 | 0.1 | 10 | 1.75 | 1 |
| CIFAR-10 | ResNet-18 (BiPer) | SGD momentum=0.9, weight decay=5e-5, lr=1e-2 | CosineAnnealingLR (eta_min=0) | 900 | 256 | 0.1 | 10 | 1.75 | 1 |
| Imagenet | ResNet-18 (BiPer) | SGD momentum=0.9, weight decay=5e-5, lr=1e-2 | CosineAnnealingLR (eta_min=0) | 300 | 512 | 0.1 | 10 | 1.75 | 1 |
| CIFAR-10 | ResNet-18 (VISPA) | SGD momentum=0.9, weight decay=1e-5, lr=0.5 | CosineAnnealingLR (eta_min=0) | 600 | 256 | 0.1 | 10 | 1.75 | 1 |
| Imagenet | ResNet-18 (VISPA) | SGD momentum=0.9, weight decay=1e-5, lr=0.5 | CosineAnnealingLR (eta_min=0) | 200 | 1024 | 0.1 | 10 | 1.75 | 1 |

# E   CUSTOM METRICS DESCRIPTION AND VALUES

Table 10: Custom metrics description.

| Name | Description | Formula (%) |
|------|-------------|-------------|
| Quality Difference | Shows how much the performance gap between the proposed model and the full-precision (FP) model has been reduced relative to the baseline. A positive value indicates improvement, while a negative value shows a drop in performance. | $\mu_q = \frac{\text{born\_result} - \text{baseline\_result}}{\text{fp\_result} - \text{baseline\_result}} \cdot 100$ |
| Relative Memory Usage | Indicates how much smaller the binarized model is compared to its FP counterpart. A lower percentage means more memory savings. | $\mu_m = 100 \left(1 - \frac{\text{binary\_weights} + 16 \cdot (\text{total\_weights} - \text{binary\_weights})}{16 \cdot \text{total\_weights}}\right)$ |
| Plateau Comparison | Measures the relative improvement in training efficiency by comparing the epoch at which training stops (plateau). A higher value indicates earlier convergence. | $\mu_p = \frac{\text{baseline\_plateau} - \text{born\_plateau}}{\text{baseline\_plateau}} \cdot 100$ |

**born_result** is the performance metric of the proposed method, **baseline_result** is the performance metric of the baseline method, **binary_weights** is total amount of binary weights, **total_weights** is total amount of weights, **baseline_plateau** is the epoch at which the performance metric reaches its maximum value for baseline method, **born_plateau** is the epoch at which the performance metric reaches its maximum value for baseline method.

Table 11: BORN Custom metrics results for image SR task on EDSR as a backbone architecture.

| Architecture | $\mu_q$ (%) | | | | $\mu_m$ (%) | $\mu_p$ (%) | | | | $\mu_q$ Mean | $\mu_m$ Mean | $\mu_p$ Mean |
|---|---|---|---|---|---|---|---|---|---|---|---|---|
| | Set5 | | Set14 | | | Set5 | | Set14 | | | | |
| | PSNR | SSIM | PSNR | SSIM | | PSNR | SSIM | PSNR | SSIM | (%) | (%) | (%) |
| EDSR (IR-Net) | 86.22 | 90.88 | 87.88 | 91.48 | 83.44 | 1.0 | 31.5 | 32.0 | 34.0 | 89.11 | 83.44 | 24.62 |
| EDSR (STE) | 55.05 | 67.06 | 54.50 | 70.89 | 83.44 | 6.1 | 33.3 | 38.1 | 36.2 | 61.88 | 83.44 | 28.43 |
| EDSR (LAB) | 24.79 | 24.19 | 26.79 | 21.15 | 83.44 | 4.0 | 5.0 | 4.0 | 10.5 | 24.23 | 83.44 | 5.875 |
| EDSR (BBCU) | 14.95 | 11.11 | 23.97 | 12.5 | 83.44 | 7.5 | 18.0 | 4.5 | 18.0 | 15.63 | 83.44 | 12.0 |
| EDSR (ReActNet) | 23.88 | 21.82 | 27.87 | 22.92 | 83.44 | 4.0 | 4.0 | 4.0 | 4.0 | 24.12 | 83.44 | 6.0 |
| **Mean EDSR Quality metrics** | **40.98** | **43.01** | **44.20** | **43.79** | **83.44** | **4.52** | **18.36** | **16.52** | **20.54** | **42.99** | **83.44** | **14.99** |

Table 12: BORN custom metrics results for image SR task on SRResNet as a backbone architecture.

| Architecture | $\mu_q$ (%) | | | | $\mu_m$ (%) | $\mu_p$ (%) | | | | $\mu_q$ Mean | $\mu_m$ Mean | $\mu_p$ Mean |
|---|---|---|---|---|---|---|---|---|---|---|---|---|
| | Set5 | | Set14 | | | Set5 | | Set14 | | | | |
| | PSNR | SSIM | PSNR | SSIM | | PSNR | SSIM | PSNR | SSIM | (%) | (%) | (%) |
| SRResNet (IR-Net) | 49.47 | 58.33 | 78.50 | 56.25 | 71.50 | 3.67 | 6.95 | 0.17 | 4.75 | 60.64 | 71.50 | 3.89 |
| SRResNet (STE) | 58.35 | 54.55 | 82.45 | 51.72 | 71.50 | 5.63 | 7.89 | 2.15 | 5.68 | 61.77 | 71.50 | 5.38 |
| SRResNet (LAB) | -4.49 | 125.00 | -360.76 | 225.00 | 71.50 | 2.35 | 29.13 | 2.40 | 10.81 | -3.81 | 71.50 | 11.17 |
| SRResNet (BBCU) | -0.69 | 0.00 | 3.45 | 0.00 | 71.50 | 7.58 | 24.71 | 24.48 | 31.31 | 0.69 | 71.50 | 22.03 |
| SRResNet (ReActNet) | 7.72 | 14.58 | 10.20 | 14.63 | 71.50 | 1.28 | 2.65 | 0.46 | 14.86 | 11.78 | 71.50 | 4.82 |
| **Mean SRResNet Quality metrics** | **22.07** | **50.49** | **-37.23** | **69.52** | **71.50** | **4.10** | **14.29** | **5.94** | **13.50** | **26.21** | **71.50** | **9.45** |

Table 13: BORN custom metrics results for LLM tasks on GPT-2 as a backbone architecture.

| Architecture | $\mu_q$ (%) | | $\mu_m$ (%) | $\mu_p$ (%) | | $\mu_q$ Mean | $\mu_m$ Mean | $\mu_p$ Mean |
|---|---|---|---|---|---|---|---|---|
| | IMDB | WikiText | | IMDB | WikiText | (%) | (%) | (%) |
| GPT-2 (IR-Net) | 46.51 | 82.9 | 64.46 | 30 | 84 | 64.705 | 64.46 | 57 |
| GPT-2 (STE) | -25.45 | 15.01 | 64.46 | -25 | 4 | -5.22 | 64.46 | -10.5 |
| GPT-2 (LAB) | 71.43 | 50.78 | 64.46 | 80 | 6 | 61.11 | 64.46 | 43 |
| GPT-2 (ReActNet) | 52.94 | 2.06 | 64.46 | 90 | 18 | 27.5 | 64.46 | 54 |
| **Mean GPT-2 Quality metrics** | **36.36** | **37.69** | **64.46** | **53.0** | **23.2** | **37.02** | **64.46** | **38.1** |

Table 14: BORN custom metrics results for image classification task on ResNet-18 as a backbone architecture.

| Architecture | $\mu_q$ (%) | | | $\mu_m$ (%) | $\mu_p$ (%) | | | $\mu_q$ Mean | $\mu_m$ Mean | $\mu_p$ Mean |
|---|---|---|---|---|---|---|---|---|---|---|
| | CIFAR-10 | Imagenet | | | CIFAR-10 | Imagenet | | | | |
| | Top-1 | Top-1 | Top-5 | | Top-1 | Top-1 | Top-5 | (%) | (%) | (%) |
| ResNet-18 (IR-Net) | 33.04 | 1.74 | 2.61 | 92.16 | 6.52 | 3.67 | 4.4 | 12.46 | 92.16 | 4.86 |
| ResNet-18 (STE) | 63.68 | 72.90 | 76.90 | 92.16 | 3.25 | 5.63 | 6.75 | 71.16 | 92.16 | 5.21 |
| ResNet-18 (LAB) | 10.54 | 1.11 | 4.76 | 92.16 | 10.37 | 2.35 | 3.05 | 5.47 | 92.16 | 5.25 |
| ResNet-18 (ReActNet) | 57.58 | 2.44 | 3.23 | 92.16 | 15 | 7.58 | 8.33 | 21.08 | 92.16 | 10.3 |
| ResNet-18 (AdaBin) | 67.39 | 4.69 | 7.41 | 92.16 | 3.11 | 6.28 | 8.79 | 26.50 | 92.16 | 6.06 |
| ResNet-18 (BiPer) | 22.52 | 0.87 | 1.98 | 92.16 | 2 | 3 | 3.6 | 8.46 | 92.16 | 2.86 |
| ResNet-18 (VISPA) | 187.50 | 1.33 | 1.90 | 92.16 | 12.45 | 1.28 | 1.92 | 63.58 | 92.16 | 5.21 |
| **Mean ResNet-18 Quality metrics** | **63.18** | **12.15** | **14.11** | **92.16** | **7.57** | **4.25** | **5.26** | **29.81** | **92.16** | **5.67** |

# F  RATIONALE FOR HYPERPARAMETERS SELECTION

In this section we provide the approach to select the values for hyperparameters used in BORN.

## F.1  SELECTION OF $T_{\min}$ AND $T_{\max}$

$T_{\min}$ and $T_{\max}$ determine the stage of the training at which the maximum value of $g'$ (see Fig 1) begins to increase. This stage can be computed as $e = \log_{\frac{T_{\max}}{T_{\min}}} \frac{1}{T_{\min}}$. For instance, selecting $T_{\min} = 1/10$ and $T_{\max} = 10$ results in $e = 1/2$, indicating that the increase starts after half of the training has been completed. Reducing $T_{\min}$ increases $e$, whereas increasing $T_{\max}$ decreases it.

## F.2  SELECTION OF $F$ AND $b$

The choice of relevant values for these parameters depends on the complexity of the backpropagation process, which is influenced by architectural and methodological factors that impact activations distribution and gradient's behavior, including: *network topology* (Vaswani et al., 2017; Krizhevsky et al., 2012), *architectural components* (He et al., 2016b; Falkena et al., 2023; Lin et al., 2015; Ioffe & Szegedy, 2015), *activation functions*, *regularization techniques* (Srivastava et al., 2014; Krogh & Hertz, 1991).

Among these factors, the variety of activation values influences the complexity of backpropagation. More diverse activation values necessitate smaller values of $b$, as a lower $b$ extends the range of nonzero gradients, allowing for more effective weight updates (see Fig 2). Moreover, the hyperparameter $F$ should not be set too high, as excessive values may lead to oscillations. Thus, tuning of $F$ and $b$ is essential to prevent instability, and optimize convergence. Also for $F$ and $b$, grid search $[1, 3]$ with a step of $1/4$ over $1/10$ of the number of epochs of the expected training showed excellent results. More advanced methods, such as Bayesian optimization or gradient-based optimization, can also be utilized for hyperparameters selection.

## F.3  SELECTION OF $\lambda_1$ AND $\lambda_2$

Parameters $\lambda_1$ and $\lambda_2$ control the influence of training progress and oscillations on weight approximation.

For $\lambda_1$ and $\lambda_2$, grid search $[0.5, 1.5]$ and $[0.125, 0.5]$ with a step of $1/8$ over $1/10$ of the number of epochs of the expected training showed excellent results. More advanced methods, such as Bayesian optimization or gradient based optimization, can also be utilized for hyperparameters selection.

Also we have performed a grid search for $\lambda_1$ and $\lambda_2$ in (11) across multiple datasets and tasks to do sensitivity analysis (see Table 15) for different pairs on ResNet-18 for both the ImageNet and CIFAR-10 datasets, showing that the performance (accuracy) is stable across a reasonable range of values — i.e., the method is not particularly sensitive to these parameters for a fixed architecture. In addition, we provide the same analysis for SRResNet (LAB and BBCU version; in field PSNR5), which, when viewed together with the ResNet-18 results, demonstrates that values are more dependent on the architecture rather than the specific dataset.

| (a) BBCU based (SRResNet) | | | |
| --- | --- | --- | --- |
| $\lambda_1/\lambda_2$ | 0.125 | 0.25 | 0.5 |
| 0.8 | 31.493 | 31.490 | 31.490 |
| 1.0 | 31.505 | 31.450 | 31.505 |
| 1.25 | 31.400 | 31.400 | 31.200 |

| (b) LAB based (SRResNet) | | | |
| --- | --- | --- | --- |
| $\lambda_1/\lambda_2$ | 0.125 | 0.25 | 0.5 |
| 0.8 | 31.461 | 31.450 | 31.450 |
| 1.0 | 31.450 | 31.430 | 31.420 |
| 1.25 | 31.420 | 31.425 | 31.100 |

| (c) ResNet-18 (CIFAR-10) | | | |
| --- | --- | --- | --- |
| $\lambda_1/\lambda_2$ | 0.125 | 0.25 | 0.5 |
| 0.8 | 91.8 | 91.7 | 90.9 |
| 1.0 | 91.87 | 91.6 | 91.4 |
| 1.25 | 91.86 | 91.2 | 90.8 |

| (d) ResNet-18 (ImageNet) | | | |
| --- | --- | --- | --- |
| $\lambda_1/\lambda_2$ | 0.125 | 0.25 | 0.5 |
| 0.8 | 62.4 | 62.5 | 62.1 |
| 1.0 | 62.6 | 62.4 | 62.3 |
| 1.25 | 62.56 | 62.3 | 62.0 |

Table 15: Sensitive analysis of $\lambda_1$ and $\lambda_2$ to architectures and datasets.

### F.4 SELECTION OF THE ADAPTIVE LR HYPERPARAMETERS

We recommend the following Adaptive LR settings (obtained via grid search). Set $\eta$ approximately equal to min_lr; use the same min_lr and max_lr values as in the baseline scheduler; avoid settings where $\eta$ is considerably smaller than min_lr (negligible effect) or much larger than min_lr (which can skew the average LR).

## G RATIONALE FOR CHOOSING tanh APPROXIMATION

In this section, we describe the rationale for choosing tanh as a sign approximation. tanh provides a trade-off between two kinds of gradient error introduced by standard Identity and Hardtanh approximations of sign (for more details, see Section 4.2 of Qin et al. (2020)). tanh approximation also has a key property — the convenient way to parametrize the gradient error with $t$ — which is sufficient for the convergence to the optimal solution of optimization problem, as was shown in Theorem 1: the more the value of $t$, the closer the weights to the optimal solution.

We also conducted a toy experiment with a fully connected network with three layers (MLP), 16 hidden neurons and different static approximations (hardtanh, identity, piecewise quadratic (Liu et al., 2018), approximations of BNN+ and BNN++ Lu et al. (2023)). We set the number of epochs equals to 400, number of iterations equals to 50 (batch size equals to 50). The training and validation data is described in Appendix B.11. We used Adam as optimizer with Cosine Annealing LR (eta_min = 1e-7), and MSE loss. For BNN+ approximation we set $\mu = 5$ (Lu et al., 2023); for BNN++ we found the best $\mu \in [5; 30] \cap \mathbb{N}$ in terms of minimal validation loss ($\mu = 6$). Table 16 contains the minimal MSE loss on the validaiton dataset for each sign approximation.

Table 16: MSE comparison of binary MLP with different sign approximations.

| Approximation | MSE loss |
| --- | --- |
| Identity | $5.3501 \times 10^{-3}$ |
| BNN+ ($\mu = 5$) | $1.9009 \times 10^{-3}$ |
| BNN++ ($\mu = 6$) | $1.7154 \times 10^{-3}$ |
| Piecewise quadratic | $1.4088 \times 10^{-3}$ |
| STE | $1.4004 \times 10^{-3}$ |
| $h\,(F = 2.5, b = 1)$ (4) | $\mathbf{4.0156 \times 10^{-4}}$ |

This experiment showed the superiority of tanh approximation. MSE loss on validation set of MLP with tanh approximation was an order of magnitude smaller than with other approximations. We will add it during the revision.

## H  NLP EXPERIMENTS SETTING

For our NLP experiments, we employ the well-established GPT-2 architecture with modifications implemented through the substitution of its binary convolution layers. We chose GPT-2 as our primary benchmark for the following reasons:

- GPT-2 (and other models of similar scale) has become the de facto intermediate-scale baseline in recent quantization studies, serving as a common reference point for comparing methods and enabling meta-analysis across works (Chitsaz et al., 2024; Park et al., 2022; Frantar et al., 2023; Xu et al., 2024; Dettmers & Zettlemoyer, 2023). Based on this experience, we can assert the applicability of our proposal for LLM using only this model. We also note that there are no works on the binarization of LLM of this size (up to a billion), which means that we are the first to propose such a benchmark, which can open up access to this area for most of the scientific community.

- GPT-2 weights are openly available and require only modest compute resources (e.g., a single high-memory GPU). This ensures that our experiments can be easily reproduced by the community without the need for large-scale clusters (Radford et al., 2019).

