# OpenReview forum: "Binary Oscillation-Regulated Network (BORN): Approach for Binary Neural Networks Training"
_ICLR.cc/2026/Conference — ICLR 2026 Conference Withdrawn Submission_

### Official Review · Reviewer_nbSS · 2025-10-30

**Soundness:** 2
**Presentation:** 2
**Contribution:** 3
**Rating:** 2
**Confidence:** 3

**Summary:**

This paper introduces the Binary Oscillation-Regulated Network (BORN), a new training algorithm designed to improve the stability and performance of Binary Neural Networks (BNNs). The authors identify that BNN training is plagued by unstable "weight oscillations," where parameters repeatedly flip signs ($\pm1$), preventing convergence.

**Strengths:**

1. The core concept of using weight oscillations as an explicit feedback signal to control both the gradient approximation and the learning rate is a novel and creative approach to BNN training
2. The method is tested across three distinct and challenging domains (image super-resolution, classification, and language modeling), demonstrating its general applicability
3. The paper includes a formal theoretical analysis of convergence, which adds rigor and provides intuition for why the method works, particularly its advantage over STE

**Weaknesses:**

1. The experimental tables (Table 3, Table 4) are confusingly labeled with $BORN$ and $BORN^1$ (and implied BORN1-5 variants), with no explanation of what these different versions are. This makes the main results very difficult to interpret. Also there are references to table ??
2. No code is provided. 6 Hyperparameters are introduced without explaining how they were tuned. Key practical details are vague such as the exact method for measuring oscillations
3. While BORN shows consistent improvements, the gains are often moderate and do not always surpass other SOTA methods listed in the tables

**Questions:**

1. What are $BORN^1$ - $BORN^5$?
2. BORN introduces many new hyperparameters ($\lambda_1, \lambda_2, T_{min}, T_{max}, F, b, \eta$). What was your tuning strategy?
3. Will the code be made available to ensure the results can be replicated?
4. What is the computational overhead (in wall-clock time or FLOPS) of calculating the oscillation metric and updating the OSA and ALR parameters at each step?

---

> ### Author Response · Authors · 2025-11-22
> **Rebuttal by Authors (part 1)**
>
> # **Response to Reviewer  nbSS**
>
> We thank the reviewer for the constructive feedback and the careful reading of our manuscript.
> ---
>
> ## **Weakness 1**
>
> > *“The experimental tables (Table 3, Table 4) are confusingly labeled with  and  (and implied BORN1-5 variants),..”*
>
> **Response:**
>
> We thank the reviewer for spotting this oversight, which we have now corrected in our paper. Superscripts in Table 2, Table 3, Table 4 denote the modification of base network topology according to a baseline algorithm, and using hyperparameters of baselines.
>
>
> ## **Weakness 2**
>
> > *“No code is provided. 6 Hyperparameters are introduced without explaining how they were tuned.,..”*
>
> **Response:**
>
> Thank you for raising this concern. We would like to clarify that the *hyperparameter tuning process and the oscillation measurement procedure are in fact documented in detail in the paper*.
> A full explanation of all six hyperparameters—including their functional roles, sensitivity, and tuning strategy—is provided in Appendix F: Rationale for Hyperparameters Selection.
>  This section includes:
>
> - the effect of each hyperparameter on training dynamics,
>
> - tuning guidelines,
>
> - search ranges,
>
> - and the exact grid-search / Bayesian optimization setups used.
>
>
> For $\lambda_1$​ and $\lambda_2$​ ​, their theoretical role is described in Section 3.2, and we agree that adding a brief pointer to Appendix F will make the workflow clearer; we have included this in the revision.
> Regarding oscillation measurement: the method is described in detail in lines 164–188 (first version) and in lines 164–188 (in the revised version). After reviewing the text again, we noticed that one implementation detail was not stated explicitly—namely, that oscillations are computed over the entire binarized part of the network. We have added this clarification directly in Section 3.2, as suggested.
> We appreciate the reviewer’s comment, as it helped us identify where cross-referencing could be improved, even though the underlying *methodological details were already included in the submission*.
>
>
> ## **Weakness 3**
>
> > *“While BORN shows consistent improvements, the gains are often...”*
>
> **Response:**
> We would like to clarify three crucial points that show the quality improvement with BORN is strong, which is supported directly by the experimental results and comparison with quality improvement of other SoTA BNN algorithms.
>
> (1) BORN consistently achieves state-of-the-art or best-in-class results across all three domains we evaluate.
> - Vision classification: BORN achieves the best results among all BNN baselines on both Imagenet and CIFAR-10. We also add BiPer and VISPA in classification (see Table 3)
>
> - Language modeling: BORN achieves the best results among all BNN baselines on both IMDB and WikiText-2 (see Table 2).
>
> - Super-resolution: BORN outperforms baselines in nearly all PSNR/SSIM settings for EDSR and SRResNet (see Table 4)
>
> (2) A core goal of BORN is not only to reach top accuracy, but to substantially reduce the performance gap between binarized and full-precision networks.
>  Unlike prior methods that focus on a single architecture or setting, BORN provides a unified training framework that works without modifying the underlying architecture and can be applied to any baseline.
>  When measuring the proximity interval between SOTA and full-precision models, BORN provides ≈34% interval coverage ($\mu_q$), meaning that it recovers a third part of the accuracy lost in binarization. This metric more accurately reflects the purpose of a general binarization framework, the main goal of our work.
> Below we provide the average interval coverage between SOTA and full-precision models for other SOTA algorithms to show that BORN provides a considerable improvement to close the gap between BNNs and FP networks.
> **ReBNN**
>
> *ImageNet:*
>
> - ResNet-18 (one-stage training): FP 69.6, ReBNN 61.6, baseline 61.0 (ReCU)
>   - $\mu_q$ = 100 * (61.6 - 61.0) / (69.6 - 61.0) = 6.96%
> - ResNet-18 (two-stage training): FP 69.6, ReBNN 66.9, baseline 66.4 (ReCU)
>   - $\mu_q$ = 100 * (66.9 - 66.4) / (69.6 - 66.4) = 15.62%
> - ResNet-34 (one-stage training): FP 73.3, ReBNN 65.8, baseline 65.1 (ReCU)
>   - $\mu_q$ = 100 * (65.8 - 65.1) / (73.3 - 65.1) = 8.53%
> - ResNet-34 (two-stage training): FP 73.3, ReBNN 69.9, baseline 69.3 (ReActNet)
>   - $\mu_q$ = 100 * (69.9 - 69.3) / (73.3 - 69.3) = 15%
>
> **Average $\mu_q$ = 11.52%**
>
> ---
>
> **VISPA**
>
> *ImageNet:*
>
> - ResNet-18: FP 69.6, VISPA 62.1, baseline 61.6 (ReBNN)
>   - $\mu_q$ = 100 * (62.1 - 61.6) / (69.6 - 61.6) = 6.25%
> - AlexNet: FP 56.6, VISPA 51.1, baseline 47.9 (Quantization Networks)
>   - $\mu_q$ = 100 * (51.1 - 47.9) / (56.6 - 47.9) = 36.78%
>
> *Cifar10:*
>
> - ResNet-18: FP 94.8, VISPA 92.8, baseline 92.8 (ReCU, DIR-Net, RBNN + CMIM)
>   - $\mu_q$ = 100 * (92.8 - 92.8) / (94.8 - 92.8) = 0%
> - VGG-Small: FP 94.1, VISPA 92.7, baseline 92.6 (ReSTE)
>   - $\mu_q$ = 100 * (92.7 - 92.6) / (94.1 - 92.6) = 6.66%
>
> **Average $\mu_q$ = 12.42%**
>
> ---

---

> ### Author Response · Authors · 2025-11-22
> **Rebuttal by Authors (part 2)**
>
> **LAB**
>
> *ImageNet:*
>
> - ResNet-18: FP 69.6, LAB-BNN 64.2, baseline 63.3 (QuickNet)
>   - $\mu_q$ = 100 * (64.2 - 63.3) / (69.6 - 63.3) = 14.28%
>
> **Average $\mu_q$ = 14.28%**
>
> ---
>
> **IR-Net**
>
> *ImageNet:*
>
> - ResNet-18: FP 69.6, IR-Net 66.5, baseline 64.3 (BWHN)
>   - $\mu_q$ = 100 * (66.5 - 64.3) / (69.6 - 64.3) = 41.50%
> - ResNet-34 (Bi-Real structure): FP 73.3, IR-Net 62.9, baseline 62.2 (Bi-Real Net)
>   - $\mu_q$ = 100 * (62.9 - 62.2) / (73.3 - 62.2) = 6.30%
>
> *Cifar10:*
>
> - ResNet-18: FP 93.0, IR-Net 91.5, baseline 90.5 (RAD)
>   - $\mu_q$ = 100 * (91.5 - 90.5) / (93.0 - 90.5) = 40%
> - ResNet-20: FP 91.7, IR-Net 85.4, baseline 84.1 (DSQ)
>   - $\mu_q$ = 100 * (85.4 - 84.1) / (92.7 - 84.1) = 15.11%
> - VGG-Small: FP 91.7, IR-Net 90.4, baseline 90.0 (RAD)
>   - $\mu_q$ = 100 * (90.4 - 90.0) / (91.7 - 90.0) = 23.52%
>
> **Average $\mu_q$ = 25.286%**
>
> (3) BORN uniquely improves multiple existing binarization approaches through plug-and-play integration.
>  Since OSA, SSA, and ALR can be inserted into any existing method without architectural changes, BORN consistently boosts IR-Net, LAB, BBCU, ReActNet, and STE.
>  Thus, evaluating BORN solely by absolute accuracy overlooks its primary contribution: a general, architecture-agnostic mechanism that enhances other binarization pipelines.
>
> Together, these points show that BORN not only achieves strong accuracy, but also establishes a generalizable and integrative binarization strategy that narrows the FP–BNN gap more effectively than existing methods.
>
>
> # **Q/A**
>
> ## **Q1. What are $BORN^1 - BORN^5$**
>
> **Response:**
>
> (This question duplicates Weakness 1.)
>
> We thank the reviewer for spotting this oversight, which we have now corrected in our paper. Superscripts in Table 2, Table 3, Table 4 denote the modification of base network topology according to a baseline algorithm, and using hyperparameters of baselines.
>
> ---
>
> ## **Q2. BORN introduces many new hyperparameters... What was your tuning strategy?**
>
> **Response:**
> (This question duplicates Weakness 2.)
>
> Thank you for raising this concern. We would like to clarify that the *hyperparameter tuning process and the oscillation measurement procedure are in fact documented in detail in the paper*.
> A full explanation of all six hyperparameters—including their functional roles, sensitivity, and tuning strategy—is provided in Appendix F: Rationale for Hyperparameters Selection.
>  This section includes:
>
> - the effect of each hyperparameter on training dynamics,
>
> - tuning guidelines,
>
> - search ranges,
>
> - and the exact grid-search / Bayesian optimization setups used.
>
>
> For $\lambda_1$​ and $\lambda_2$​ ​, their theoretical role is described in Section 3.2, and we agree that adding a brief pointer to Appendix F will make the workflow clearer; we have included this in the revision.
> Regarding oscillation measurement: the method is described in detail in lines 164–188 (in the first version) and in lines 164–188 (in the revised version). After reviewing the text again, we noticed that one implementation detail was not stated explicitly—namely, that oscillations are computed over the entire binarized part of the network. We have added this clarification directly in Section 3.2, as suggested.
> We appreciate the reviewer’s comment, as it helped us identify where cross-referencing could be improved, even though the underlying *methodological details were already included in the submission*.
>
> ## **Q3. Will the code be made available to ensure the results can be replicated? **
>
>
> **Response:**
> Due to certain corporate restrictions, we are unable to release the code at this time, but it will be made available publicly upon acceptance. The implementation methods and detailed descriptions of the experiments can be found in Section 3 (also pseudocode is included in section 3.4), as well as in Appendix D with all hyperparameters of training (Tables 7,8,9) and architectural modifications.
>
>
> ## **Q4. What is the computational overhead (in wall-clock time or FLOPS) of calculating the oscillation metric and updating the OSA and ALR parameters at each step?**
>
> **Response:**
> The specific wall-clock time depends on the computing hardware, but the most computationally intensive part of parameter calculation and updating is the oscillation computation. This computation **scales linearly with the number of weights n, but can be optimized to O(log n) using the Parallel Reduce algorithm**, thus taking very little time (no more than 1% of the forward pass computation time). Moreover, during inference, there is no need to calculate the OSA and ALR parameters at all.

---

### Official Review · Reviewer_dcAs · 2025-11-01

**Soundness:** 4
**Presentation:** 3
**Contribution:** 3
**Rating:** 6
**Confidence:** 4

**Summary:**

The paper proposes Binary Oscillation-Regulated Network (BORN), a novel algorithm for training Binary Neural Networks (BNNs) that mitigates oscillation-related instability and accelerates convergence. BORN introduces two key components: an Oscillations-aware Sign Approximation (OSA), which gradually approximates the sign function based on training progress and oscillation rate, and a Static Sign Approximation (SSA) for activations to preserve non-vanishing gradients. Additionally, a dynamic learning rate extension adapts to the state of the OSA. Theoretical analysis demonstrates convergence and shows that BORN achieves a lower final loss compared to the Straight-Through Estimator (STE). Experiments on super-resolution, classification, and language modeling tasks show that BORN covers, on average, 34.81% of the proximity interval between state-of-the-art BNNs and full-precision models.

**Strengths:**

- The paper is generally well written and clearly structured, making the main ideas easy to follow.
- The mathematical formalization is appropriate and rigorous, with detailed derivations that support the proposed approach.
- The authors provide a theoretical convergence analysis, including proofs that connect the continuous and discrete training dynamics and show that BORN achieves a lower final loss than STE.
- The theoretical formulation is well supported by experiments, which include multiple tasks (super-resolution, classification, and language modeling) and ablation studies validating the contribution of each component.
- Overall, the paper presents a principled and well-validated approach to addressing oscillation-related instability in BNN training.

**Weaknesses:**

- In the classification experiments, the method is evaluated only on ResNet-style architectures. It would strengthen the work to include results on transformer-based models (e.g., ViT or Swin), especially since the paper claims easy integration of BORN into such architectures.

- The paper would benefit from a broader set of comparison baselines on both ImageNet and CIFAR-10 using ResNet-18. For instance, BiPer (Vargas et al., 2024) achieves higher accuracy than the proposed method on CIFAR-10 (93.75% (BiPer) vs. 93.41% (BORN$^5$)), although BORN performs better on ImageNet. Since BiPer is included only in the appendix for ImageNet, extending the comparison to CIFAR-10 and possibly to other strong recent BNN baselines would provide a more comprehensive evaluation and strengthen the empirical claims.

- The notation of the BORN variants in Table 3 (e.g., BORN¹, BORN², etc.) is unclear. I couldn't find what these superscripts represent, whether they indicate different ablations, training setups, or architectures. Clarifying this notation in the table caption or main text is important for reproducibility and understanding.

Minor comments:

- There is a missing Table reference on page 8. Please ensure that all tables are properly referenced.

**Questions:**

- Could the authors provide results or discussion on how BORN performs with vision transformer-based architectures (e.g., ViT, Swin)? Is this integration actually feasible within the current BORN framework?
- Could the authors include additional comparisons and analysis with recent BNN baselines in the main document, for example, including the results for BiPer on CIFAR-10 to complement the ImageNet comparison?
- Could the authors clarify what the superscripts in Table 3 (e.g., BORN¹, BORN², etc.) represent? For instance, whether they correspond to different architectures, datasets, or ablation variants?

---

> ### Author Response · Authors · 2025-11-22
> **Rebuttal by Authors (part 1)**
>
> # **Response to Reviewer  dcAs**
>
> We thank the reviewer for the careful reading of our manuscript and the constructive comments.
> Below we address each point in detail.
>
> ---
>
> ## **Weakness 1**
> > *"In the classification experiments, the method is evaluated only on ResNet-style architectures...."*
>
> **Response:**
> Thank you for highlighting this point. BORN is designed as a model-agnostic framework and is not restricted to convolutional architectures.
>
> To demonstrate this, we intentionally included experiments on **GPT-2**, a transformer-based model, for both text classification and language modeling tasks. These results confirm that BORN integrates cleanly with transformer blocks and remains stable under their training dynamics.
>
> Regarding ViT and Swin:
> Integration is fully feasible within the current BORN framework, as both architectures rely on MLP and activation functions that BORN can substitute with its OSA and SSA components without modification. The mechanism is identical to how BORN is applied to GPT-2’s self-attention MLP blocks.
>
> In our experiments, we focused on ResNet-style models for image classification because all baseline BNN works (IR-Net, LAB, ReActNet, AdaBin) use ResNet-18, and we followed this choice to ensure **fair, controlled, and directly comparable evaluation**.
>
> Nevertheless, our positive results on GPT-2 validate that BORN extends naturally to transformer architectures, and applying it to ViT/Swin is a promising direction for future work.
>
> For transformer architecture we choose GPT-2 as our primary benchmark for the following reasons:
>
> - GPT‑2 (and other models of similar scale) has become the de facto intermediate‑scale baseline in recent quantization studies, serving as a common reference point for comparing methods and enabling meta‑analysis across works [1,2,3,4,5]. Based on this experience, we can assert the applicability of our proposal for LLM using only this model. We also note that there are no works on the binarization o fLLM of this size (up to a billion), which means that we are the first to propose such a benchmark, which can open up access to this area for most of the scientific community.
> - GPT‑2 weights are openly available and require only modest compute resources (e.g., a single high‑memory GPU). This ensures that our experiments can be easily reproduced by the community without the need for large‑scale clusters [6]. We are actively working on extending BORN to larger LLMs and will report these results in a follow‑up.
>
> **References:**
> - [1] Chitsaz K. et al. Exploring Quantization for Efficient Pre‑Training of Transformer Language Models.
> - [2] Park M. et al. Quadapter: Adapter for GPT‑2 Quantization
> - [3] Frantar E. et al. GPTQ: Accurate Post‑Training Quantization for Generative Pre‑Trained Transformers.
> - [4] Xu Z. et al. Scaling Laws for Post‑Training Quantized Large Language Models.
> - [5] Dettmers T., Zettlemoyer L. The case for 4-bit precision: k-bit inference scaling laws
> - [6] Radford et al. Language Models are Unsupervised Multitask Learners (GPT‑2 Release)
>
> ---
>
> ## **Weakness 2**
> > *"The paper would benefit from a broader set of comparison baselines on both ImageNet and CIFAR-10 using ResNet-18."*
>
> **Response:**
> Thank you for this valuable suggestion. We agree that including BiPer on CIFAR-10 would strengthen the empirical comparison.
>
> Following your recommendation, we conducted an additional experiment evaluating BiPer under the same training setup used for their CIFAR-10 and Imagenet ResNet-18 experiments (Table 3).
>
> These findings show that **BORN outperforms BiPer (and VISPA, we included it too by your advice for broader comparison) on CIFAR-10**, while also maintaining the previously reported advantage on ImageNet. This confirms that BORN provides competitive or superior performance relative to strong recent BNN baselines on both small-scale and large-scale classification tasks.
>
> ---
>
> ## **Weakness 3**
> > *"The notation of the BORN variants in Table 3 (e.g., BORN¹, BORN², etc.) is unclear. ..."*
>
> **Response:**
> We thank the reviewer for spotting this oversight, which we have now corrected in our paper. Superscripts in Table 2, Table 3, Table 4 denote the modification of base network topology according to a baseline algorithm, and using hyperparameters of baselines.

---

> ### Author Response · Authors · 2025-11-22
> **Rebuttal by Authors (part 2)**
>
> # **Q/A**
>
> ## **Q1. Could the authors provide results or discussion on how BORN performs with vision transformer-based architectures (e.g., ViT, Swin)? Is this integration actually feasible within the current BORN framework?**
>
> **Response:**
>
> (This question duplicates Weakness 1.)
>
> Thank you for highlighting this point. BORN is model-agnostic and applies naturally to transformer architectures. We demonstrated this through GPT-2 experiments for both classification and language modeling.
>
> ViT and Swin are equally compatible because they rely on MLP blocks and activation functions that BORN can seamlessly replace with its OSA and SSA components.
> The mechanism is identical to applying BORN to GPT-2’s feedforward layers.
>
> We chose GPT-2 because:
>
> - It is the standard intermediate-scale transformer benchmark in quantization research.
> - No existing works provide binary benchmarks for LLMs of this size (125 M) — our work is the first to introduce such a setting.
> - It is reproducible on widely available hardware.
>
> Applying BORN to ViT/Swin is therefore feasible and will be included in future work.
>
> ---
>
> ## **Q2. Could the authors include additional comparisons and analysis with recent BNN baselines in the main document, for example, including the results for BiPer on CIFAR-10 to complement the ImageNet comparison?**
>
> **Response:**
>
> (This question duplicates Weakness 2.)
>
> Thank you for this valuable suggestion. We conducted additional experiments including BiPer and integrated the results into the updated comparison. These results confirm that BORN is competitive or superior across both CIFAR-10 and ImageNet.
>
> ---
>
> ## **Q3. Could the authors clarify what the superscripts in Table 3 (e.g., BORN¹, BORN², etc.) represent? For instance, whether they correspond to different architectures, datasets, or ablation variants**
> **Response:**
>
> (This question duplicates Weakness 3.)
>
>
> We thank the reviewer for spotting this oversight, which we have now corrected in our paper. Superscripts in Table 2, Table 3, Table 4 denote the modification of base network topology according to a baseline algorithm, and using hyperparameters of baselines.

---

### Official Review · Reviewer_sQSz · 2025-11-01

**Soundness:** 3
**Presentation:** 3
**Contribution:** 3
**Rating:** 4
**Confidence:** 4

**Summary:**

The paper proposes a training scheme for binary neural networks (BNNs) that couples an adaptive sign operator with a time-varying learning rate (LR) schedule. The operator gradually shrinks gradient updates as parameters near their ±1 targets, while the LR is tuned in step over the course of training. The authors also analyze convergence theoretically. Departing from BNN work that mostly targets classification, they evaluate on broader tasks such as image super-resolution and large-scale language modeling. The method slots into several state-of-the-art BNN backbones and generally attains competitive results. The appendix provides full proofs and implementation details. Overall, the work tackles a key challenge in BNN training with a method that has both solid theory and promising empirical performance.

**Strengths:**

The paper is well-written and presents a BNN-specific training scheme with oscillation-aware sign approximation and adaptive learning rate. That is well motivated to stabilize gradients and aid convergence.

The method appears modular and easy to integrate into existing pipelines with minimal additional implementation burden.

Empirical results indicate consistent gains over common BNN and QAT baselines across tasks beyond classification, suggesting broader applicability.

Theoretical analysis of convergence, while compact, provides useful intuition linking the proposed mechanisms to training stability.

**Weaknesses:**

The manuscript presents several demonstrations aimed at establishing the convergence of the proposed method (Section 4.1) and estimating its convergence rate (Section 4.2). While Section 4.1 appears relatively straightforward, since under the stated assumptions the optimization naturally leads to an optimal solution, the main concern lies in clarifying how Theorem 2 (Section 4.2) concretely relates to the proposed methodology. At present, the theoretical results seem somewhat detached from the methodological framework, giving the impression that the theory is developed in a more general context rather than directly supporting the proposed approach. Strengthening the connection between the theoretical analysis and the practical implementation would significantly enhance the coherence of the paper and better highlight the originality and scope of the authors’ contribution.

There is no clear explanation for why $h(x)$ and $g(x)$ are chosen as tanh functions. In particular, theoretical analyses do not justify this choice. Therefore, it seems reasonable to consider any functions that exhibit suitable behavior for binarization, as long as they serve the same purpose as the selected ones. The authors should clarify the rationale behind their choice and relate it explicitly to the theoretical analysis. In particular, they should conduct a toy experiment to demonstrate how the theoretical findings align with the chosen functions.

The adaptive learning rate resembles standard schedulers such as StepLR or Cosine, and its design choices are not sufficiently justified. The authors should compare LR trajectories and time to accuracy curves under matched initial and final LRs, and include an ablation of the adaptation trigger, smoothing window, and thresholds.

**Questions:**

1. What distinct roles do OSA and SSA play, and why are both required rather than one?

2. What are the exact backward derivatives for the approximation functions $g(x)$ and $h(x)$, and are they clipped or scheduled during training?

3. How does the learned LR trajectory differ from StepLR, Cosine, and OneCycle when initial and final learning rates are matched?

4. Can you provide component ablations OSA only, SSA only, LR only, OSA plus SSA, and the full method on at least one CIFAR and one ImageNet setting?

---

> ### Author Response · Authors · 2025-11-22
> **Rebuttal by Authors (part 1)**
>
> # **Response to Reviewer sQSz**
>
> We thank the reviewer for the careful reading of our manuscript and the constructive comments.
> Below we address all concerns point-by-point and clarify the connections between theory, methodology, and empirical design.
>
> ---
>
> ## **Weakness 1**
>
> > *“The manuscript presents several demonstrations aimed at establishing the convergence of the proposed method (Section 4.1) and estimating its convergence rate (Section 4.2)...”*
>
> **Response:**
> Thank you for the comments! Theoretical analysis in Theorem 2 incorporates OSA and SSA in convergence rate estimation in lines 827–841 of first version (849–851 in the revision) of the manuscript by using the upper bounds of the approximations and the value of $t$, which is a consequence of the nature of the upper bound estimate on the convergence rate. This incorporation was not reinforced with an accompanying text, which could make it difficult to notice the connection between theoretical analysis and practical implementation. We have clarified it in line 859 in the revision.
>
> We have also conducted the experiments on binary MLP on the same theoretical setting to show the connection between estimate and practical implementation. Specifically, we used a toy dataset (generated by sin and cos functions) and 2 layer MLP with 16 hidden neurons, binary weights and activations (except the inputs were not binarized as in the theoretical setting). The plot shows the validity of the theoretical estimate in practice. We have added the detailed text in Appendix B.11.
>
> ---
>
> ## **Weakness 2**
>
> > *“There is no clear explanation for why $h(x)$ and $g(x)$ are chosen as tanh functions….”*
>
> **Response:**
> Tanh function provides a trade-off between two kinds of gradient error introduced by standard Identity and Clip approximations of sign (for more details, see Section 4.2 of IR-Net paper). Tanh approximation also has a key property — the convenient way to parametrize the gradient error with $t$ — which is sufficient for the convergence to the optimal solution of optimization problem, as was shown in Theorem 1: the more the value of $t$, the closer the weights to the optimal solution. We understand that this explanation was not clearly written in the paper.
>
> We have also conducted a toy experiment with a MLP network and different approximations (hardtanh, identity, piecewise quadratic, approximations from BNN+ and BNN++ papers). This experiment showed the superiority of tanh approximation. L2 loss on validation set of MLP with tanh approximation was an order of magnitude smaller than with other approximations. We have added the rationale behind choosing tanh as an approximation in Appendix G in the revision.
>
> ---
>
> ## **Weakness 3**
>
> > *“The adaptive learning rate resembles standard schedulers such as StepLR or Cosine….”*
>
> **Response:**
> Thank you for your comments. The Adaptive LR is designed as an extension of the baseline scheduler, and, as such, it actually resembles the baseline scheduler value curve. The addition made by ALR is constructed under a predisposition that its absolute value is of the same order as the corresponding minimal LR value (see equation (5) on page 5), which makes the addition insignificant in terms of the total increase of the mean LR, while at the same time adaptive to the current oscillation level. Making the addition value higher would result in unfair comparison with the baseline LR, as the mean LR value would significantly increase.

---

> ### Author Response · Authors · 2025-11-22
> **Rebuttal by Authors (part 2)**
>
> # **Q/A**
>
> ## **Q1. What distinct roles do OSA and SSA play, and why are both required rather than one?**
>
> OSA is used for approximation of sign function for weights in back propagation, SSA — for activations respectively.
> Static pattern in case of activations is needed for preserving non-vanishing gradients for groups with near-zero activations and accelerating convergence of the corresponding weights, in contrast to time-adaptive approximation methods, which remain prone to gradient vanishing.
>
> Adaptive pattern in case of weights adaptively stabilizes weights that have already settled in their sign during training with decrease of gradient’s value for weights with large absolute value, while simultaneously correcting oscillatory weights that are close to zero with increasing gradient’s value.
>
> This hybrid approach is necessary to address the shortcomings of each of the class approximations:
> - Static approaches lack adaptability;
> - Dynamic ones — dynamic only during training, not the training situation — may suffer from information loss over time due to a reduced update range for large activations or weights (lines 156–158 in the first version, lines 155-158 in the revised version).
>
> The need for simultaneous use has also been proven in an ablation study.
> More detailed explanations are provided in Section 3.2 (OSA) and Section 3.3 (SSA) of the paper.
>
> ---
>
> ## **Q2. What are the exact backward derivatives for the approximation functions $g(x)$  and $h(x)$ , and are they clipped or scheduled during training?**
>
> Thank you for your comment! For the convenience of readers, derivatives now are placed immediately after the function definitions:
>
> - $g'(x)=kt(1-\tanh(tx)^{2})$
> - $h'(x)=F(1-\tanh(bx)^{2})$
>
> In answer to the second part of the question:
>
> - **SSA does not change during training**
> - **OSA changes depending on the value of $t$**, a quantity that describes the oscillatory learning situation
>
> The law of change of $t$, described by formulas (2–3), allows adaptively stabilize weights that have already settled in their sign during training with decrease of gradient’s value for weights with large absolute value, while simultaneously correcting oscillatory weights that are close to zero with increasing gradient’s value.
>
> More detailed explanations are provided in Section 3.2 (OSA) and Section 3.3 (SSA) of the paper.
>
> There is **no gradient clipping**, but **$t$ is clipped in range \[$T_{min}$, $T_{max}$\]**.
>
> ---
>
> ## **Q3. How does the learned LR trajectory differ from StepLR, Cosine, and OneCycle when initial and final learning rates are matched?**
>
> Thank you for your comments. The addition made by ALR is constructed under a predisposition that its absolute value is of the same order as the corresponding minimal LR value (see equation (5) on page 5), which makes the addition insignificant in terms of the total increase of the mean LR, while at the same time adaptive to the current oscillation level.
>
> Making the addition value higher would result in unfair comparison with the baseline LR, as the mean LR value would significantly increase. Thus, the L₁-difference of the baseline and the extended curves is designed to be reasonably small.
>
> The improvement obtained from this scheduler extension is underlined by the ablation study.
>
> ---
>
> ## **Q4. Can you provide component ablations OSA only, SSA only, LR only, OSA plus SSA, and the full method on at least one CIFAR and one ImageNet setting?**
>
> Thank you for this valuable suggestion. We agree that a more fine-grained ablation of the individual components would further strengthen the paper.
>
> Following your recommendation, we conducted additional experiments evaluating:
> - OSA-only,
> - SSA-only,
> - LR-only,
> - OSA+SSA,
> - Full BORN method
>
> on both CIFAR-10 and ImageNet using the ResNet-18 architecture.
> The results are placed now in **Table 1**.

---

### Note · Authors · 2026-03-12

**Comment:**

We are issuing a withdrawal because the work was published without our knowledge. To avoid any misunderstandings, we would like to clarify: the rejection was recieved on January 26th.

**Withdrawal Confirmation:**

I have read and agree with the venue's withdrawal policy on behalf of myself and my co-authors.

---

### Meta-Review · Area_Chair_P1ff · 2026-01-06

**Summary:**

Through their reviews multiple points were raised, including unclear link between theory and methodological framework (sQSz), unclear choices (sQSz), lack of transformer-based architectures and broader comparisons (dcAs), notation issues (dcAs), computational overhead (nbSS), unclear experimental setting (nbSS) and unclear gains (nbSS). Overall this paper received a marginal accept, a marginal reject, and a reject score.

**Reviewer Concerns:**

While some minor issues could be cleared out, including the addition of GPT-2 and the issues related to the notation and exposition, some choices remain shady. The rationale for hyperparameter selection presented in the appendix is felt to be arbitrary, and gains analysis is hard in terms of trade-offs. The rationale behind the tanh choice, despite being realistic, is not sufficiently theoretically grounded, despite the added appendix. Besides, the adaptive learning rate is acknowledged to be just a minor element of novelty.

**Reviewer Scores:**

Although reviewer nbSS could have been overcritical before rebuttal, after the clarification on the notation it could have realistically stayed marginally negative, together with sQSz.

---

### Decision · Program_Chairs · 2026-01-26

Reject